# Iterative SuFEx approach for sequence-regulated oligosulfates and its extension to periodic copolymers

Min Pyeong Kim[1], Swatilekha Kayal[1], Chiwon Hwang[2], Jonghoon Bae[3], Hyunseok Kim [4], Dong Gyu Hwang[1], Min Ho Jeon[5], Jeong Kon Seo[3], Dowon Ahn[2], Wonjoo Lee[2], Sangwon Seo [6], Joong-Hyun Chun[5] ✉, Youngchang Yu [2] ✉ & Sung You Hong [1] ✉

The synthesis of sequence-regulated oligosulfates has not yet been established due to the difficulties in precise reactivity control. In this work, we report an example of a multi-directional divergent iterative method to furnish oligo-sulfates based on a chain homologation approach, in which the fluorosulfate unit is regenerated. The oligosulfate sequences are determined by high resolution mass spectrometry of the hydrolyzed fragments, and polysulfate periodic copolymers are synthesized by using oligomeric bisfluorosulfates in a bi-directional fashion. The synthetic utility of this iterative ligation is demonstrated by preparing crosslinked network polymers as synthetic adhesive materials.

Iterative synthetic approaches have been widely employed for the chemical synthesis of diverse sequenced molecules, including biologically relevant molecules and artificial synthetic compounds, via the stepwise elongation of specific building blocks[1–5]. Iterative ligation methods for accessing oligonucleotides, oligopeptides, and oligosaccharides typically employ successive unmasking and coupling steps to assemble their respective monomeric building blocks (e.g., protected monosaccharides, amino acids, and nucleotides)[6–11]. In the iterative synthesis of non-natural compounds, monodisperse polyesters have also been prepared by using a protection and deprotection strategy[12]. In particular, an iterative binomial synthetic method has enabled the exponential growth of a molecular assembly[13,14]. Moreover, iterative synthetic methods have also been used to afford higher-generation divergent dendritic molecules[15,16], and chain homologation-type reactions have been recently developed to afford polyethers and polyketides via 1,2-metallate rearrangements[17,18]. Despite these tremendous developments, the use of heteroatom-based molecular linkages has been relatively

underexplored compared to C–C, C–O, or C–N bond-based ligation methods[19].

Sulfur(VI) fluoride exchange (SuFEx) has been regarded as another ideal click chemistry to give sulfate-based linkages due to its modularity, orthogonal reactivity, and synthetic competence[20–27]. SuFEx coupling reactions typically operate between SuFEx substrates (e.g., sulfonyl fluorides, sulfamoyl fluorides, and aryl fluorosulfates) and readily available aryl alcohols, silyl ethers, or amines under nitrogenous Lewis base catalysts[28–33]. Recently, sequenced oligomers and polymers have been synthesized by repetitive SuFEx coupling and copper(I)-catalyzed azide–alkyne cycloaddition (CuAAC) reactions (Fig. 1a)[34]. The multi-component reaction approach using SuFEx coupling, CuAAC, and Ugi reactions to give higher-order structures was also reported[35,36]. Despite these advances, the formation of sequence-regulated oligosulfates via the iterative assembly of SuFExable units remains a challenge.

Since 2014, there has been significant progress on the use of SuFEx chemistry to create new polymers, including polysulfates,

[1]Department of Chemistry, Department of Chemical Engineering, and Graduate School of Carbon Neutrality, Ulsan National Institute of Science and Technology (UNIST), Ulsan 44919, Republic of Korea. [2]Center for Advanced Specialty Chemicals, Korea Research Institute of Chemical Technology (KRICT), Ulsan 44412, Republic of Korea. [3]UNIST Central Research Facility (UCRF), UNIST, Ulsan 44919, Republic of Korea. [4]Department of Chemistry, Pohang University of Science and Technology (POSTECH), Pohang 790-784, Republic of Korea. [5]Department of Nuclear Medicine, Yonsei University College of Medicine, Seoul 03722, Republic of Korea. [6]Department of Physics and Chemistry, Daegu Gyeongbuk Institute of Science and Technology (DGIST), Daegu 42988, Republic of Korea. ✉e-mail: jchun@yuhs.ac; ycyu@krict.re.kr; syhong@unist.ac.kr

**a** SuFEx and click-based iterative approach

*Niu - 2018*

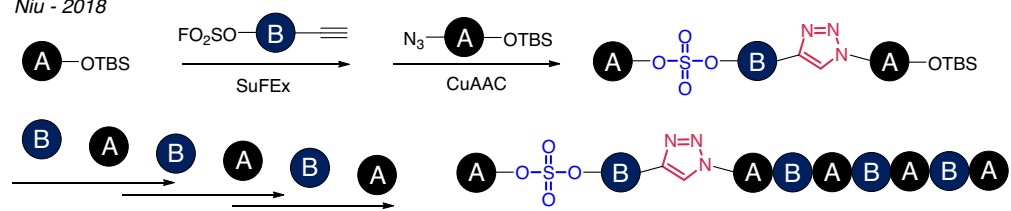

**b** Polysulfate alternating and block copolymerization

*Sharpless & Fokin - 2014*　　　　　　　　*Sharpless & Wu - 2021*

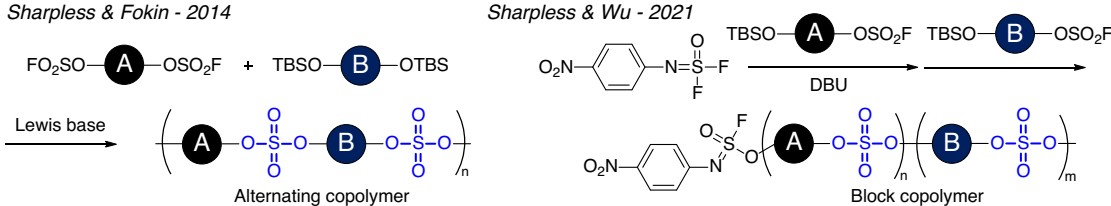

**c  This work: divergent iterative synthesis of oligosulfates and its polymerization**

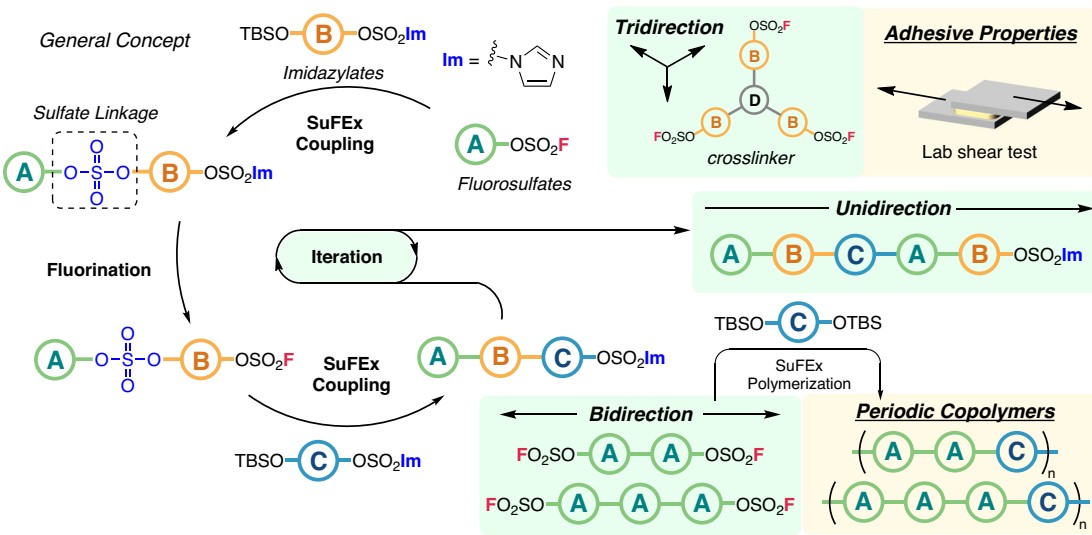

**Fig. 1 | Overview of sequence-regulated iterative synthesis. a** SuFEx-based sequence-regulated polymers and oligomers. **b** Copolymerization of polysulfates. **c** Synthesis of sequenced-regulated oligosulfates and its extension to synthesis of polysulfate periodic copolymers and adhesives.

polysulfamides, and polysulfluoridoimidates[37–44]. In particular, polysulfates have been prepared through the step-growth polycondensation reactions between bis(aryl fluorosulfates) and silyl-protected diol monomers to give homo- and alternating copolymers (Fig. 1b)[37,38,40,44]. More recently, a chain-growth polycondensation method has been reported to furnish block copolymer structures[45]. However, facile synthetic routes to afford sequence-regulated backbone structures and polysulfate periodic copolymers have not yet established, primarily due to difficulties in achieving the precise control of reactivity.

In the present work, a divergent iterative SuFEx approach to yield sequence-regulated oligosulfates that are solely constructed from SuFExable units is reported (Fig. 1c). This iterative elongation is accomplished by the homologative regeneration of fluorosulfates via the nucleophilic fluorination of an imidazylate intermediate. Thus, the alternation of SuFEx coupling and nucleophilic fluorination allows the precise regulation of sulfate linkages via the controlled assembly of building blocks in both uni-, bi-, and tri-directional manners. The key challenge in this chain homologation approach is that the imidazylates must be unreactive toward the SuFEx coupling. In addition, this

bi-directional elongation is applied to the synthesis of polysulfate periodic copolymers. The synthetic utility is then highlighted by the preparation of three-dimensional network polymers, which are examined by lab shear tests for adhesive applications.

## Results

### Orthogonality of the reaction conditions

Recently, we reported the en route synthesis of aryl fluorosulfates (ArOSO$_2$F) from phenols having isolable aryl imidazylate intermediates (ArOSO$_2$Im)[46]. Based on the work, it was anticipated that the corresponding fluorosulfates from imidazylates by nucleophilic fluorination can facilitate homologous chain elongation in case imidazylates remain intact upon SuFEx coupling. At this outset, the SuFEx coupling reaction between 3,5-dimethylphenyl fluorosulfate and bisphenol A-derivative bearing silyl ether- and imidazylate moieties was investigated (Supplementary Information (SI), Section 2.1). Bisphenol-based building blocks were selected due to their rigidity and structural diversity, as reflected by their importance in the production of engineered polymers[47,48]. Encouragingly, the result indicated that the imidazylate functional group was tolerable to the SuFEx coupling

conditions in the presence of 1,8-diazabicyclo[5.4.0]undec-7-ene (DBU) as a Lewis base catalyst, while giving the product in 95% yield. Subsequently, after optimization of the fluorination reaction (SI, Section 2.2, Supplementary Table 2), the fluorosulfate group was regenerated by treatment with AgF to furnish a homologated fluorosulfate in 88% yield, which can be attributed to the potential chain homologation-type growth.

## Stepwise growth of sequence-regulated oligosulfates

After establishing the reactions conditions in hand, the iterative synthesis was initiated from bisphenol A-derivative **1** bearing a methoxy masking moiety for uni-directional growth. Then, bisphenol E-derivative **2** bearing silyl ether and imidazylate moieties was coupled to **1** to give dimeric bisphenol derivative of imidazylate **3**. This was then fluorinated using AgF to afford fluorosulfate homolog **4** while

preserving the sulfate linkage. Next, the second SuFEx coupling reaction was performed with dimeric **4** and monomeric **5** in the presence of DBU to synthesize the trimeric structure **6** bearing an imidazylate moiety. Eventually, repeated cycles of nucleophilic fluorination and SuFEx coupling gave the sequence-regulated oligosulfates pentamer (Fig. 2a). The three distinct bisphenol-derived monomeric units **1**, **2**, and **5** (symbolized as {A}, {B}, or {C} in Fig. 2a) were used. These are distinguished by their characteristic $^1H$ NMR peaks at ca. 1.6 (s, 6H, 2xCH$_3$), 1.5 (d, $J$ = 7.2 Hz, 3H, CH$_3$), and 4.0 ppm (s, 2H, CH$_2$), respectively. This made it possible to monitor the sequential oligomeric growth via the appearance or increased integration values of these peaks. In addition, despite some peak overlaps, the $^{13}C$ NMR signals of the aliphatic and aromatic carbons could also be used to evaluate the stepwise growth. Notably, the SuFEx coupling reaction was confirmed by the appearance of the imidazylate peak at 7.8 ppm in the $^1H$ NMR

**Fig. 2 | Uni-directional iterative synthesis of sequence-regulated oligosulfates.** **a** Synthetic scheme of uni-directional iteration from **1** to **11**, alternating SuFEx coupling reaction and nucleophilic fluorination. **b** Synthesized uni-directional sequence-regulated oligosulfates. Reaction conditions: (i) fluorosulfates (1.0 equiv), TBS/imidazylate building block (1.0 equiv), DBU (25 mol %), MeCN, 80 °C, 2 h; (ii) imidazylate (1.0 equiv), AgF (1.6 equiv), MeCN, 80 °C, 16 h.

spectrum. Likewise, the appearance of the characteristic peak at 36 ppm in [19]F NMR spectrum indicated the formation of the fluorosulfate functional group by nucleophilic fluorination. In total, the five distinct sequence-regulated oligosulfate pentamers were synthesized and characterized, including MeO–{**A-B-C-A-B**}–OSO$_2$Im (**11**), MeO–{**A-C-C-B-B**}–OSO$_2$Im (**18**), MeO–{**A-A-C-B-C**}–OSO$_2$Im (**25**), MeO–{**A-B-A-C-B**}–OSO$_2$Im (**30**), and MeO–{**A-C-B-A-B**}–OSO$_2$Im (**35**) (Fig. 2b). Remarkably, the iteration of SuFEx coupling and nucleophilic fluorination maintained high yields, making this a reliable synthetic protocol.

## Bi-directional growth of sequence-regulated oligosulfates

We next shifted our attention from unidirectional iterative synthesis to bi-directional growth. In this case, bisfluorosulfate-bearing starting materials, such as 4,4′-(propane-2,2-diyl)bis(4,1-phenylene) difluorosulfate (**36**), were used. Thus, the bisfluorosulfate **36** was reacted with **2** to afford trimeric bisimidazylate **38**, which was then fluorinated to give trimeric bisfluorosulfate **39** (Fig. 3a). By alternating these reactions, the bi-directional growth with odd-numbered sequencing units was achieved to synthesize bisfluorosulfate pentamer **41**. In addition, the growth having even-numbered sequencing units was established by the synthesis of dimeric imidazylate **42** by the SuFEx coupling reaction of monomers **3** and **37**. Following this iterative approach, the effective bi-directional synthesis using oligosulfates **43** afforded tetrameric bisfluorosulfate **45** (Fig. 3b). Consequently, bi-directional growth was achieved successfully in both even and odd number increment to obtain the following four sequence-regulated bisfluorosulfates: FO$_2$SO–{**C-B-A-B-C**}–OSO$_2$F

(**41**), FO$_2$SO–{**A-C-C-A**}–OSO$_2$F (**45**), FO$_2$SO–{**A-A**}–OSO$_2$F (**47**), FO$_2$SO–{**A-A-A**}–OSO$_2$F (**49**) (Fig. 3c).

## Determination of oligosulfate sequences through hydrolyzed fragments

We then investigated the direct analytic method to read the oligosulfate sequences. It was reported that diaryl sulfates can be hydrolyzed under acidic or basic conditions, or deblocked by iodides and azides, and polysulfates can be degraded under basic conditions using DBU, BTMG, KOH or ammonia solution[45,49,50]. Referring to these, a fragmentation test was conducted to detect the expected fragments (see SI: Supplementary Table 3 and Section 5.1). Then, two hydrolysis reaction conditions were established: (i) NaOH at 80 ˚C in DMF, and (ii) NaOH at room temperature in a DCM/MeOH mixture (Fig. 4a). We chose **11** and **30** as model compounds because they have the same molecular weights in the two different pentamer sequences of {**A-B-C-A-B**} and {**A-B-A-C-B**}. Interestingly, these two pentamers exhibited different hydrolysis patterns (Figs. 4b and 4c). In the case of **11**, the four fragments were detected, namely: MeO–{**A**}–OH, MeO–{**A-B**}–OH, MeO–{**A-B-C**}–OH, and MeO–{**A-B-C-A**}–OH. However, in the case of **30**, the four observed fragments were MeO–{**A**}–OH, MeO–{**A-B**}–OH, MeO–{**A-B-A**}–OH, and MeO–{**A-B-A-C**}–OSO$_2$OH. In addition, the isotopic patterns of the corresponding fragments were investigated by high-resolution mass spectrometry (HRMS) (SI, Section 5.2).

## Application to polysulfate periodic copolymers

While investigating the bi-directional synthetic approach, it was speculated that SuFEx coupling between the bisfluorosulfates

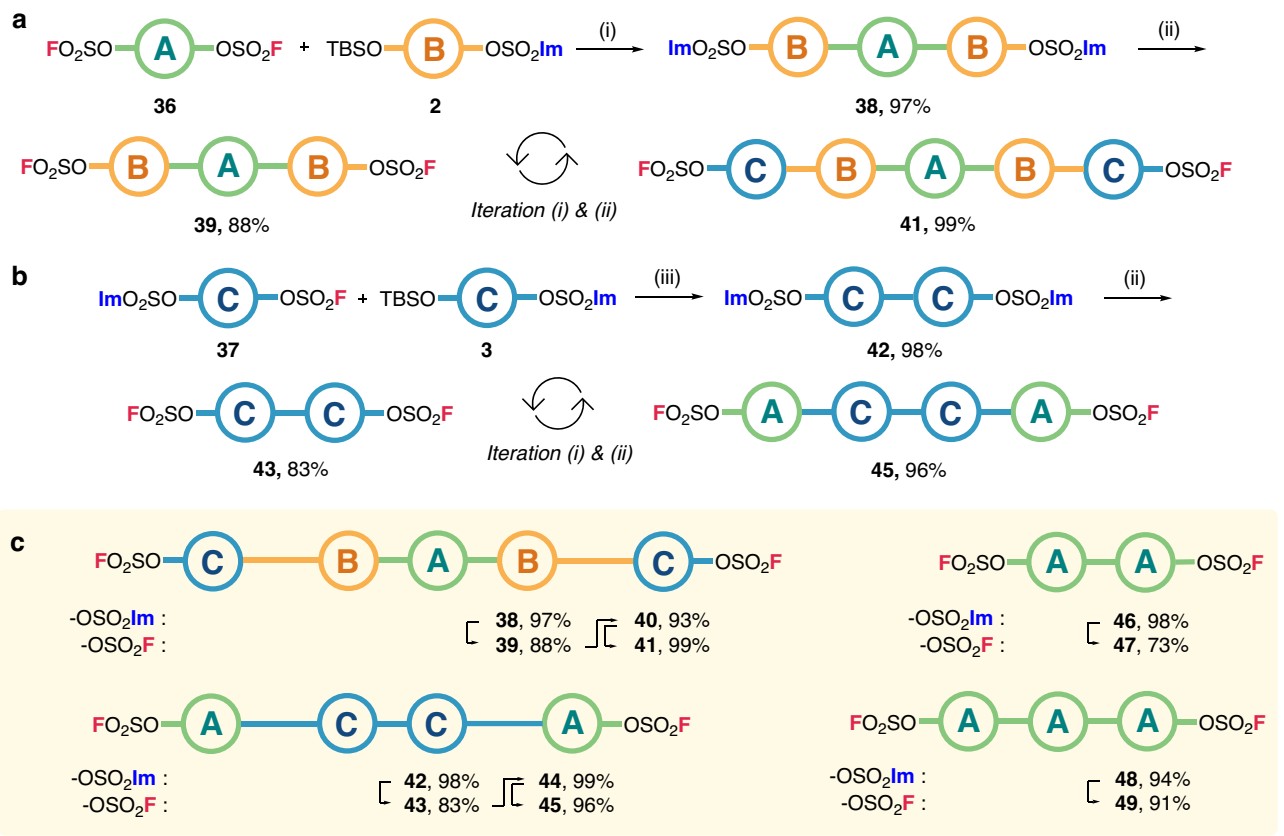

**Fig. 3 | Bi-directional iterative synthesis of sequence-regulated oligosulfates.** **a** Synthetic strategy of odd-numbered bi-directional oligomers. **b** Iterative elongation of even-numbered oligomeric bisfluorosulfates. **c** Synthesized bi-directional sequence-regulated oligosulfates. Reaction conditions: (i) bisfluorosulfate (1.0 equiv), TBS/imidazylate building block (2.0 equiv), DBU (25 mol %), MeCN, 80 ˚C, 2 h; (ii) bisimidazylate (1.0 equiv), AgF (3.0 equiv), MeCN, 80 ˚C, 16 h; (iii) imidazylate/fluorosulfate building block (1.0 equiv), TBS/imidazylate building block (1.0 equiv), DBU (25 mol %), MeCN, 80 ˚C, 2 h.

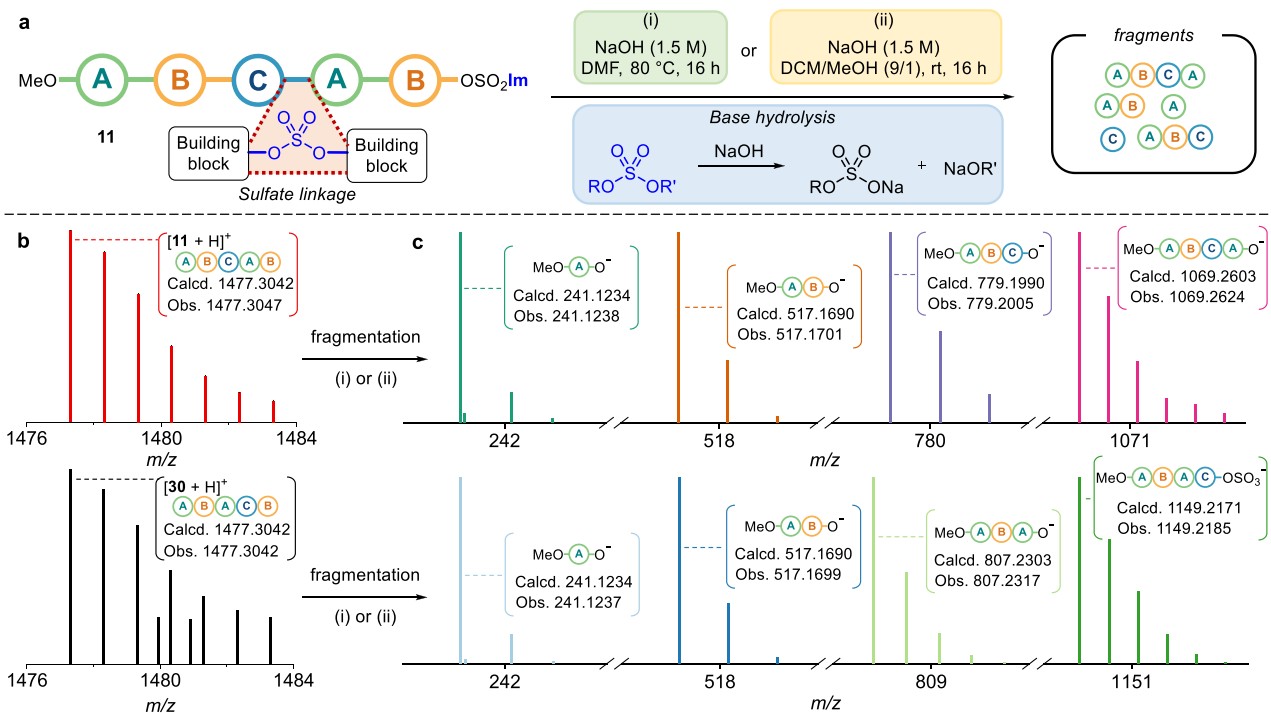

**Fig. 4 | Oligosulfates sequencing by HRMS analysis. a** Optimal conditions on hydrolysis of oligosulfates. **b** HRMS (ESI + ) data of **11** (up) and **30** (down) in centroid mode. **c** Oligomer sequencing by hydrolyzed fragments from **11** (up), and **30** (down) through HRMS (ESI-) analysis in centroid mode.

oligomers **45**, **47**, and **49** and the bis-silyl ether bearing monomers **50** and **51** might yield polysulfate periodic copolymers. Hence, three distinct polysulfate periodic copolymers of -(**A-A-C**)$_n$-, -(**A-A-A-C**)$_n$-, and -(**A-A-A-C-C**)$_n$-, designated **P-1**, **P-2**, and **P-3**, respectively, were synthesized (Fig. 5a). Even oligomeric bisfluorosulfates were used, the polycondensation yields were over 74%. The gel permeation chromatography (GPC) analysis equipped with three polystyrene-gel columns in series gave $M_n$ values of 19 kDa ($Đ = 1.5$), 24 kDa ($Đ = 1.7$), and 48 kDa ($Đ = 2.0$) for **P-1**, **P-2**, and **P-3**, respectively. Further, [1]H NMR analysis revealed the characteristic peaks of the dimethyl group within monomeric unit {**A**} at ca. 1.6-1.7 ppm and the benzylic group within unit {**C**} at 4.0 ppm. The relative integration values of these peaks within the polymers were in accordance with the structures of **P-1**, **P-2**, and **P-3**. As a result, in comparison to previously reported polysulfates affording block copolymers and alternating copolymers[43–45], our method efficiently provides polysulfate periodic copolymers, even when oligomeric bisfluorosulfates are used as monomers.

In addition, the depolymerization reactions of polysulfate periodic copolymers were conducted via base hydrolysis (Fig. 5b). The resulting GPC chromatogram revealed the change in molecular weight under the hydrolysis conditions, with the unimodal peak of **P-2** being shifted to the longer retention time, lower molecular weight region after base hydrolysis (Fig. 5c). And similar behaviors were observed for the other polysulfate copolymers, **P-1** and **P-3** (SI, Section 7.1). Then, we were curious about the depolymerization patterns of polysulfates. Therefore, the depolymerized fragments from **P-2** were analyzed by HRMS and several fragment peaks of HO-{**A**}-OSO$_2$OH, HO-{**C**}-OSO$_2$OH, HO-{**C-A**}-OH, HO-{**A-A**}-OH, and HO-{**A-A-A**}-OH were obtained from the crude mixture after the hydrolysis (Fig. 5d). These results indicated that the polysulfate linkages were generally broken into alcoholic and sulfuric acid fragments (SI, Section 7.2).

## Adhesive strength test using elongated trifluorosulfate

After developing chain homologation approach, we looked into the applicability of our method to produce functional materials. SuFEx

chemistry has exhibited versatile synthetic applications across various fields, encompassing medicinal chemistry[51,52], radiochemistry[53,54], and bioconjugation[55,56]. Particularly noteworthy is the expanding utilization of SuFEx-based functional materials in recent years[57–61]. Specifically, the use of polysulfates has shown promise in thin film capacitors, delivering high energy densities[62]. In this study, we delve into the adhesive properties of the synthetic SuFEx polymers. The linear polysulfates from the bisfluorosulfate **36** and bis-silyl ether **51** monomers exhibited that inferior adhesive strength from initial test (SI, Section 8). It is typically more advantageous to have three-dimensional network forms rather than linear polymers to achieve stronger lap shear strength[63,64]. Hence, the three-dimensional network structure obtained via SuFEx polymerization was investigated with the aim of enhancing the mechanical strength of the adhesive. Therefore, the elongated trisfluorosulfate **54** was prepared through subsequent fluorination of imidazylate **53**, which originated from triphenylmethyl-cored **52** (Fig. 6a). The adhesive resin **P-4** was synthesized by using 20 mol % DBU as a catalyst and 2:3 molar ratio of **55**: **36**, with 30 min of pre-polymerization at 130 °C, while the adhesive resin **P-5** was prepared from **54** and **51** under similar conditions, but with only 5 min of pre-polymerization at 160 °C (Fig. 6b).

The resulting pre-polymerized adhesive resins were then evenly applied to stainless steel (SUS) specimens, and cured for 2 h at 160 °C in a vacuum oven (Fig. 6b). As a result, adhesive sample **P-4** from lab shear tests by universal testing machine exhibited maximum load and lap shear strength of 1556.32 N and 2.41 MPa, respectively, along with a tensile extension at breaking point of approximately 0.31 mm (Fig. 6c). Meanwhile, the adhesive sample **P-5**, which was prepared using elongated trisfluorosulfate **54**, exhibited maximum load and lap shear strength of 1706.13 N and 2.64 MPa, respectively, along with a tensile extension of approximately 0.34 mm, as shown in Fig. 6c. The higher lap shear strength of the **P-5** adhesive compared to the **P-4** may originate from the lower self-steric hindrance of **P-5** due to the bisphenolic derivative of **54**[63–65]. Also, the higher extension of the **P-5** may be

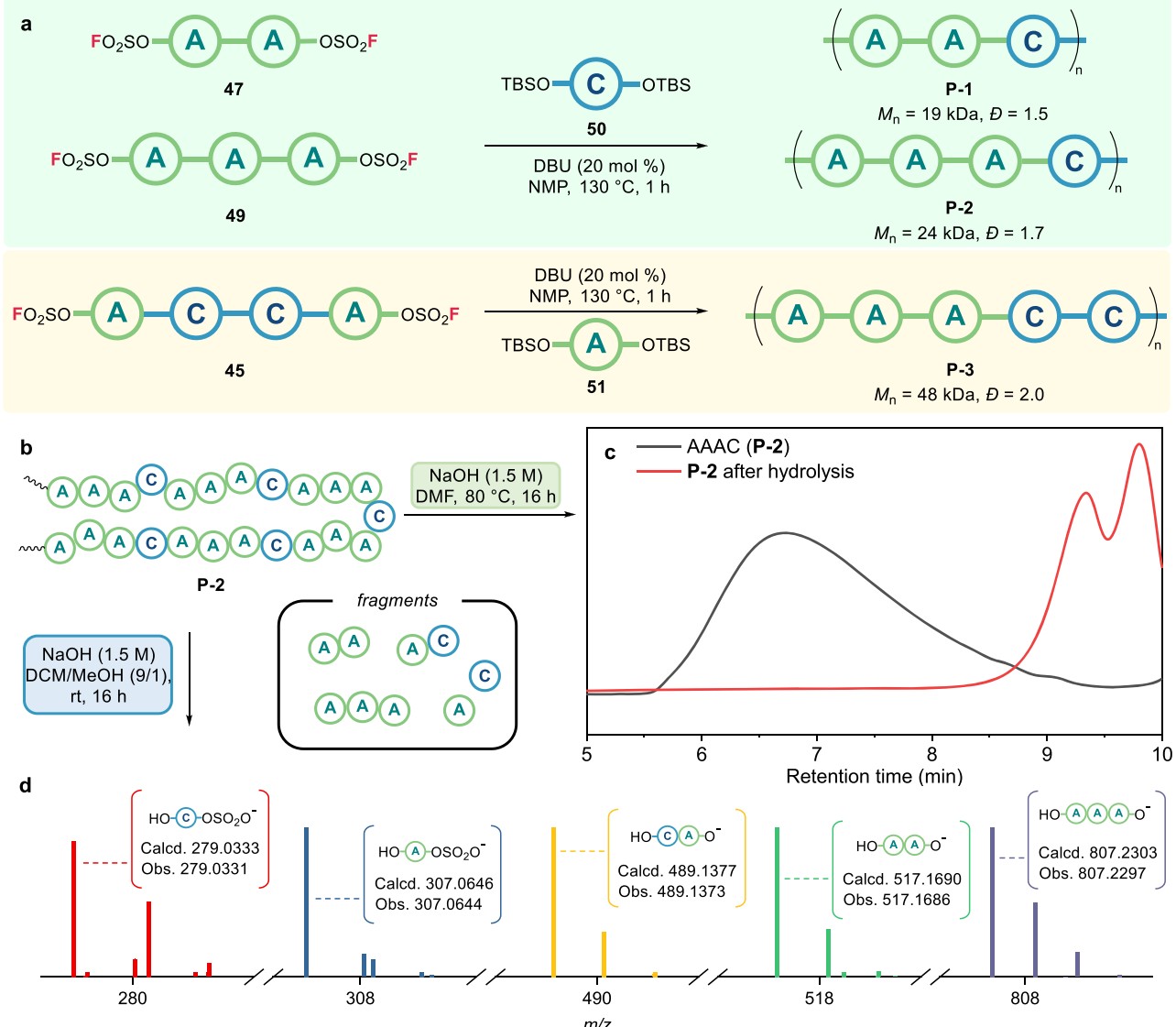

**Fig. 5 | Synthesis and analysis of polysulfates periodic copolymers. a** Synthesis of polysulfates periodic copolymers using bi-directional oligosulfates **45**, **47**, and **49**. GPC analysis via three-column method. **b** Degradation of polysulfates periodic copolymers by base hydrolysis. **c** GPC chromatogram via one-column method of before (black) and after (red) hydrolysis of **P-2**. **d** HRMS (ESI-) data of hydrolyzed fragments from **P-2** in centroid mode.

due to the lower crosslinking density resulting from the modification with the bisphenolic derivative of **54**[66,67]. Additionally, the improved adhesive characteristics can be attributed to the low viscosity of the resin generated after pre-polymerization, which facilitates better wetting of the substrate. As a result, the adhesive samples were able to withstand a 12 kg kettlebell, as shown in Fig. 6d.

In conclusion, we have demonstrated the iterative synthesis of sequence-regulated oligosulfates through successive cycles of SuFEx coupling and nucleophilic fluorination reactions. This chain homologation approach permitted the precise modulation of sequences, lengths, and directions of iterative elongation. The resulting oligosulfate sequences were determined by HRMS analysis of the hydrolyzed fragments. In addition, the bi-directional iterative growth approach for the bisfluorosulfates was expanded to the synthesis of polysulfate periodic copolymers. Tri-directional approach using elongated trisfluorosulfates was used for network polymers having the improved adhesive properties. We believe that this iterative methodology can contribute to make homologous substances with fluorosulfate moieties. Moreover, we envisage that modulating of sequences within polysulfates can allow fine-tuned properties of well-defined polymeric materials.

## Methods

### General procedure for SuFEx coupling reaction

A mixture of aryl fluorosulfate, *tert*-butyldimethylsilyl ether with imidazylate moiety, and 1,8-diazabicyclo[5.4.0]undec-7-ene (DBU) in anhydrous MeCN was stirred for 2 h at 80 °C in an Ace pressure tube. Then, the mixture was concentrated in vacuo, and purified by flash column chromatography to afford the corresponding oligosulfate-bearing imidazylate functional group.

### General procedure for nucleophilic fluorination

A mixture of imidazylate and silver(I) fluoride in anhydrous MeCN was stirred for 16 h at 80 °C in an Ace pressure tube. Then, the mixture was concentrated in vacuo, and purified by flash column chromatography to afford the corresponding oligosulfate-bearing fluorosulfate functional group. Full experimental details are found in the Supplementary Methods.

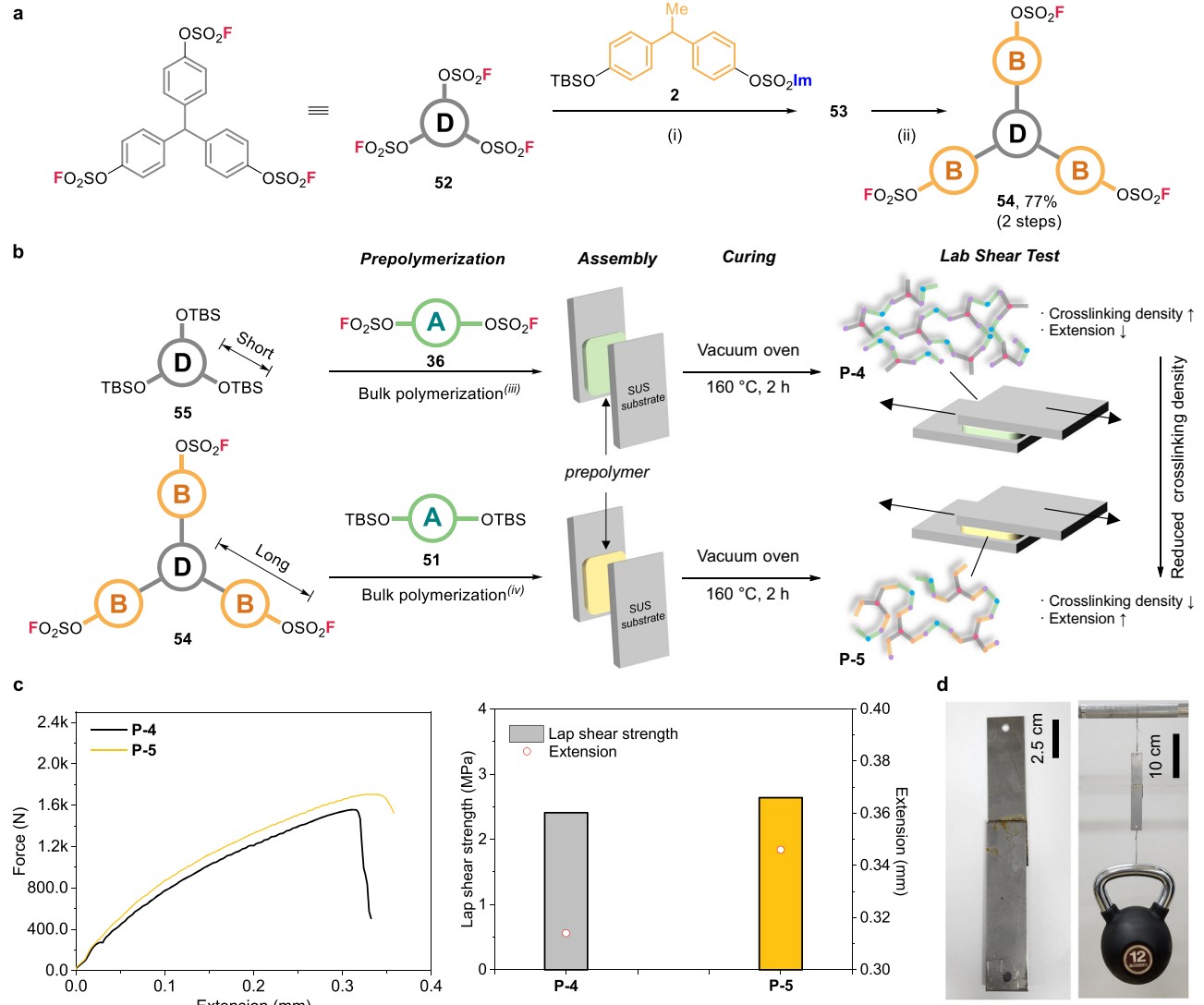

**Fig. 6 | Tri-directional synthesis and lab shear strength test of network polymers. a** Tri-directional synthesis of elongated trisfluorosulfate from core tris-fluorosulfate. **b** Schematic illustration on stepwise fabrication of adhesives **P-4** and **P-5** using **55** and **54**. **c** The results of force−extension curves and lap shear strength of SuFEx adhesives. **d** Photographs of SuFEx adhesive **P-5** bonded to the substrates to lift a 12 kg kettlebell. Reaction conditions: (i) **52** (1.0 equiv), **2** (3.3 equiv), DBU (35 mol %), MeCN, 80 ˚C, 2 h; (ii) **53** (1.0 equiv), AgF (4.5 equiv), MeCN, 80 ˚C, 16 h; (iii) **55** (1.0 equiv), **36** (1.5 equiv), DBU (20 mol %), neat, 130 ˚C, 30 min.; (iv) **54** (1.0 equiv), **51** (1.5 equiv), DBU (20 mol %), neat, 160 ˚C, 5 min.

## Data availability
All data provided in this work are available within the article and Supplementary Information files or from the corresponding author.

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

## Acknowledgements

The authors thank the support by the National Research Foundation of Korea (NRF) grant funded by the Korea government (MSIT) (NRF-2020M3H4A3081883, NRF-2020R1A5A1019631, NRF-2023R1A2C1004563, and NRF-2022R1A2C2004064), by Korea Institute of Energy Technology Evaluation and Planning (KETEP) grant funded by the Korea government (MOTIE) (20215610100040, Development of 20 Wh seawater secondary battery unit cell), and by Korea Research Institute of Chemical Technology (KS2341-10).

## Author contributions

M.P.K., J.-H.C., Y.Y. and S.Y.H. conceived the project. M.P.K., S.K. and D.G.H. synthesized compounds. M.P.K. carried out polymerization. M.P.K. and J.B. worked on oligosulfate sequencing. M.P.K. and C.H. fabricated adhesives. M.P.K. characterized compounds and polymers by NMR, GPC, and HRMS and analyzed data. C.H. and Y.Y. analyzed the data of lab shear test. M.P.K., J.-H.C., Y.Y. and S.Y.H. wrote the manuscript. M.P.K., S.K., C.H., J.B., H.K., D.G.H., M.H.J., J.K.S., D.A., W.L., S.S., J.-H.C., Y.Y. and S.Y.H discussed the results and edited the manuscript.

## Competing interests

The authors declare no competing interests.
