## [Peer Review File · Nature Communications]

Iterative SuFEx approach for sequence-regulated oligosulfates and its extension to periodic copolymersReviewers' Comments:

Reviewer #1:

Remarks to the Author:

Polysulfates and polysulfonates show excellent materials properties, including toughness and impact resistance, making them attractive for numerous applications. This manuscript represents a significant contribution to this rapidly evolving and impactful field. Specifically, it describes a novel method for synthesizing sequence-defined oligosulfates by leveraging iterative transformations of imidazylate temporary activating groups to fluorosulfates. The authors initially demonstrated the tolerability of the imidazylate functional group under SuFex coupling conditions. They then optimized the reaction conditions to convert it into a fluorosulfate group after treatment with AgF, allowing it to readily participate in the next iterative cycle. The successful implementation of the optimized conditions led to the synthesis of sequence-controlled oligosulfates in a stepwise and/or bidirectional growth manner. The significance of this methodology lies in its potential to manufacture polymers with precise control over substitution patterns and length. This capability enables the modulation of material properties with a high degree of precision. The authors finally demonstrated the fabrication of three-dimensional networks, having a trifluorosulfate as an anchor, to attain stronger lap shear strength. The mechanical properties of the polymers were calculated, revealing variations depending on the substitution pattern. The supplementary information provided by the authors is excellent and well-structured. Lastly, the references are meticulously cited, encompassing all the topics discussed in the paper.

Given the functional potential of the chemical matter that can now be accessed in a much more efficient and modular fashion, I would recommend this for publication in Nature Communication if the following comments/questions are addressed:

1. This methodology fundamentally depicts the linear growth of oligosulfate chains by interconverting a functional group. To pursue a multi-directional approach, it would necessitate an additional sulfate activating group that is orthogonal to the imidazylate intermediate. This orthogonal group would enable the selective functionalization of one direction over another, providing complete control over the pattern. Therefore, I suggest reconsidering the use of the term "multi-directional" and exploring alternative phrasing.
2. While the authors extensively discuss previous advancements in the production of such materials, they neglect to address their applicability. It is advisable to integrate relevant text and corresponding references into the manuscript to underscore the significance and practical applications of these materials.
3. Maintain consistent terminology, particularly in reference to the captions of the figures (e.g., Fig1. Sequence-regulated and sequence-controlled, etc.), for coherence.
4. In certain ¹H NMR spectra (e.g., S5, S9, etc.), an additional signal around 2.14 ppm is observed without a specific assignment. If this signal does not correspond to the product, it should be indicated with an arrow on the spectra.

Reviewer #2:

Remarks to the Author:

Review of Kim et al – Nature Comm 2024 Iterative SuFEx approach for sequence-regulated oligosulfates ..

Kim et al have provided a nice paper outlining the synthesis of sequence-regulated oligosulfates. Following (quoted) work of e.g. Niu (ACIE 2018) who pioneered the use of SuFEx towards sequence-defined oligomers, the authors noted that this had not been done for oligosulfates yet. The key of the SuFEx reaction as a click reaction is that the S-F bond does not react with almost anything else apart from O and N-centered nucleophiles, and the list of tolerated groups is in this paper extended with sulfur imidazylates. [Although this is THE key moiety of the entire paper, the authors do not give its

structure, neither in the paper nor anywhere in the supporting information.. One is obliged to look at their previous Org Lett 2020 paper = ref 44 to be elucidated.]

The key issue is evaluate the novelty of this paper is 2-fold: is this surprising, and do the authors do something truly novel with it?

The answer to the first is: more yes than no, but not fully clear. The imidazole moiety is a relatively strong electron-donating group -> to replace it in a bi-functional molecule outfitted with both an S-F and an S-Im bond with an oxygen-centered nucleophile, any reaction at the S-Im site will be outcompeted by the F-Si bond-formation catalysis of the SuFEx variant chosen by the authors. In other words: the S-Im bond will be stable, and as the authors show it is basically orthogonal to the silyl-protected SuFEx variant. I might (and perhaps: would) have guessed the first, but that the authors were able to show that it is truly orthogonal and that it can be used to form even more complex structures in typical click yields, I would not have guessed -> with some consideration, I thus think they take the "Nat Comm-novelty hurdle" here.

And do they do something novel with this? This is a resounding yes: the structures have not been made before, and they display the variation that one would expect for a high-level paper! The structures shown in Figures 1-3 are simply beautiful. They will make many people in the field think: wow, I wish my group would have (been able to) do(ne) that! The authors do not only make 'pretty pictures', but the Supp Info provides the supporting data with convincing clarity -> really nice stuff! After concluding the paper should be published in Nature Comm the question is: can it be/does it need to be improved? Yes, also here a resounding yes.

Apart from outlining to the broad readership of this journal what 'Im' is, I would suggest to improve the manuscript as follows before acceptance:

1) Polysulfates have been shown to be superior materials for thin-film capacitors (Sharpless, Wu, Joule 2023). Pls add this ref., as it puts the usefulness of these materials in a wider context.

2) The paper presents many references to older reviews of SuFEx – given the rapid development of this field, likely any reviews prior to 2021 are becoming superfluous to introduce people to the field, as e.g. shown by the involvement of Sharpless himself in e.g. the Nat -series 2023 review. [= ref 20]

3) The key feature is the stability of the S-Im group under SuFEx conditions. Yet, no clarity is provided on why that is indeed the case. The paper would thus be helped significantly by some quantum chemistry that calculated the transition states and driving forces of both reactions (for reaction at the S-F side and at the S-Im side), to provide insight in the chemoselectivity rather than just observations.

4) Polycondensation reactions are expected to give polymers with dispersity \bar{D} 'somewhat' smaller than 2.0. This is what is also observed for all SuFEx-based polymers up to now (papers by Dong, Wu, Sharpless, Zuilhof, etc.) Yet, the current authors report \bar{D} values of 1.03 and 1.06 for two of their three polymers. These numbers deserve some more attention, as also proven by their broad GPC traces, which are more in line with previous literature data, and less with mono-disperse polymers -> I would suggest the authors to reinvestigate and recalculate these, as there likely is no monodispersity observed for their polymers (in all honesty: this is also not expected, and nothing against the quality of the paper).

5) The authors artificially divide the paper in a Results & Discussion section, but in truth their Results contain some discussion (as it should: there are no uninterpreted data), and their Discussion is 'just' a Conclusion section under a different name. I would suggest to rename them as such.

6) It is already known that the SuFEx linkages are stable under neutral and acidic conditions, and break down under basic conditions. The authors also do this, and present their case, by using 0.30 mmol in 0.1 mL solutions (these conditions are only mentioned to those readers stubborn enough to continue searching the Supp Info to p S27). This is thus a 3 M (!) NaOH solution... -> almost any heterolytic bond will break down by heating overnight under these conditions.. The authors do also not compare their results with the more gentle breakdown of ref 36, which I think presented the first breakdown of SuFEx-based polymers, nor study the development of the macromolecular mass over time in neutral media at/near R.T. (which is what is most interesting in view of any applications). Here some more experimental work would surely improve the paper further.

Finally, it was a joy to read the paper – it is clear, specific enough to the specialist yet (apart from the 'missing-Im' enticing enough to draw also non-specialists to this field -> great fit for Nature Comm, I

think!

[My name is Han Zuilhof - I have no objection of presenting my name to the authors.]

Reviewer #3:

Remarks to the Author:

In this paper, Kim et al. reported an iterative method for the synthesis of oligosulfates based on a chain homologation approach, in which the fluorosulfate was regenerated by treating imidazylate intermediate with AgF, a methodology developed by part of the authors before. Polysulfate periodic copolymers are also synthesized from oligomeric bisfluorosulfates. Although several new oligosulfates and polysulfate periodic copolymers were made, it is more like short multiple-step synthesis with an imidazylate as the protecting group plus reliable SuFEx reactions. From the practical perspective, many other regular protecting groups, which could be deprotected in excellent yields before the re-installation of a fluorosulfate to one side of the molecule in one pot, might be more suitable for this task. Overall, I think the scientific contributions of this paper do not meet the high-quality requirements of Nat. Commun. Thus I don't support the publication of the paper in this Journal.

Manuscript Type: Research Article

Title: Iterative SuFEx approach for sequence-regulated oligosulfates and its extension to periodic copolymers

REVIEWER COMMENTS

Reviewer #1 (Remarks to the Author):

Polysulfates and polysulfonates show excellent materials properties, including toughness and impact resistance, making them attractive for numerous applications. This manuscript represents a significant contribution to this rapidly evolving and impactful field. Specifically, it describes a novel method for synthesizing sequence-defined oligosulfates by leveraging iterative transformations of imidazylate temporary activating groups to fluorosulfates. The authors initially demonstrated the tolerability of the imidazylate functional group under SuFEx coupling conditions. They then optimized the reaction conditions to convert it into a fluorosulfate group after treatment with AgF, allowing it to readily participate in the next iterative cycle. The successful implementation of the optimized conditions led to the synthesis of sequence-controlled oligosulfates in a stepwise and/or bidirectional growth manner. The significance of this methodology lies in its potential to manufacture polymers with precise control over substitution patterns and length. This capability enables the modulation of material properties with a high degree of precision. The authors finally demonstrated the fabrication of three-dimensional networks, having a trisfluorosulfate as an anchor, to attain stronger lap shear strength. The mechanical properties of the polymers were calculated, revealing variations depending on the substitution pattern. The supplementary information provided by the authors is excellent and well-structured. Lastly, the references are meticulously cited, encompassing all the topics discussed in the paper.

Given the functional potential of the chemical matter that can now be accessed in a much more efficient and modular fashion, I would recommend this for publication in Nature Communication if the following comments/questions are addressed:

Response) We express our gratitude to the reviewer for providing feedback on our manuscript. Below, we have included detailed responses addressing each of the points you raised in your comments.

Q1) This methodology fundamentally depicts the linear growth of oligosulfate chains by interconverting a functional group. To pursue a multi-directional approach, it would necessitate an

additional sulfate activating group that is orthogonal to the imidazylate intermediate. This orthogonal group would enable the selective functionalization of one direction over another, providing complete control over the pattern. Therefore, I suggest reconsidering the use of the term "multi-directional" and exploring alternative phrasing.

Response) As suggested by the reviewer, we have reconsidered and replaced the term of ‘multi-directional’ with ‘divergent’ in the main text, to encompass the concept of uni-, bi-, or tri-directional growth. The changes have been highlighted in yellow.

Q2) While the authors extensively discuss previous advancements in the production of such materials, they neglect to address their applicability. It is advisable to integrate relevant text and corresponding references into the manuscript to underscore the significance and practical applications of these materials.

Response) We have incorporated the following sentences into the main text to discuss the recent applications of organosulfates.

(Main text, page 6, highlighted in yellow): SuFEx chemistry has exhibited versatile synthetic applications across various fields, encompassing medicinal chemistry^{51,52}, radiochemistry^{53,54}, and bioconjugation^{55,56}. Particularly noteworthy is the expanding utilization of SuFEx-based functional materials in recent years^{57–61}. Specifically, the use of polysulfates has shown promise in thin film capacitors, delivering high energy densities⁶². In this study, we delve into the adhesive properties of the synthetic SuFEx polymers.

Newly added references:

(Main text, Ref. 51). Brighty, G. J. et al. Using sulfuramidimidoyl fluorides that undergo sulfur(VI) fluoride exchange for inverse drug discovery. *Nat. Chem.* **12**, 906–913 (2020)

(Main text, Ref. 52). Liu, Z. et al. SuFEx click chemistry enabled late-stage drug functionalization. *J. Am. Chem. Soc.* **140**, 2919–2925 (2018).

(Main text, Ref. 53). Kim, M. P. et al. Direct ¹⁸F-fluorosulfurylation of phenols and amines using an [¹⁸F]FSO₂⁺ transfer agent generated in situ. *J. Org. Chem.* **88**, 6263–6273 (2023).

(Main text, Ref. 54). Zheng, Q. et al. Sulfur [¹⁸F]fluoride exchange click chemistry enabled ultrafast late-stage radiosynthesis. *J. Am. Chem. Soc.* **143**, 3753–3763 (2021).

(Main text, Ref. 55). Zhao, S., Zeng, D., Wang, M. & Jiang, X. C-SuFEx linkage of sulfonimidoyl fluorides and organotrifluoroborates. *Nat. Commun.* **15**, 727 (2024).

(Main text, Ref. 56). Liu, F. et al. Biocompatible SuFEx click chemistry: thionyl tetrafluoride (SO₂F₄)-derived connective hubs for bioconjugation to DNA and proteins. *Angew. Chem. Int. Ed.* **58**, 8029 (2019).

(Main text, Ref. 57). Moon, H. et al. Elastic Interfacial layer enabled the high-temperature performance of

lithium-ion batteries via utilization of synthetic fluorosulfate additive. *Adv. Funct. Mater.* **33**, 2303029 (2023).

(Main text, Ref. 58). Wan, H. et al. SuFEx-enabled elastic polysulfates for efficient removal of radioactive iodomethane and polar aprotic organics through weak intermolecular forces. *Angew. Chem. Int. Ed.* **61**, e202208577 (2022).

(Main text, Ref. 59). Subramaniam, M., Ruggeri, F. S. & Zuilhof, H. Degradable click-reaction-based polymers as highly functional materials. *Matter* **3**, 2490–2492 (2022).

(Main text, Ref. 60). Kassick, A. J., Chen, L., Kovaliov, M., Mathers, R. T., Locklin, J. & Averick, S. SuFEx-based strategies for the preparation of functional particles and cation exchange resins. *Chem. Commun.* **55**, 3891–3894 (2019).

(Main text, Ref. 61). Xia, C. et al. Tunable electronic memory performances based on poly(triphenylamine) and its metal complex via a SuFEx click reaction. *Chem. Asian J.* **14**, 4296–4302 (2019).

(Main text, Ref. 62). Li, H. et al. High-performing polysulfate dielectrics for electrostatic energy storage under harsh conditions. *Joule* **7**, 95–111 (2023).

Q3) Maintain consistent terminology, particularly in reference to the captions of the figures (e.g., Fig1. Sequence-regulated and sequence-controlled, etc.), for coherence.

Response) In response to your suggestion, we have amended the term ‘sequence-controlled’ to revised to ‘sequence-regulated’ to maintain the consistency in the main text (Figs. 1 and 3).

Fig. 1 | Overview of sequence-regulated iterative synthesis. **a** SuFEx-based sequence-regulated polymers and oligomers. **b** Copolymerization of polysulfates. **c** Synthesis of sequence-regulated oligosulfates and its extension to synthesis of polysulfate periodic copolymers and adhesives.

Fig. 3 | Bi-directional iterative synthesis of sequence-regulated oligosulfates. **a** Synthetic strategy of odd-numbered bi-directional oligomers. **b** Iterative elongation of even-numbered oligomeric bisfluorosulfates. **c** Synthesized bi-directional sequence-regulated oligosulfates.

Q4) In certain ¹H NMR spectra (e.g., S5, S9, etc.), an additional signal around 2.14 ppm is observed without a specific assignment. If this signal does not correspond to the product, it should be indicated with an arrow on the spectra.

Response) The resonance observed at approximately δ 2.14 ppm in the ¹H NMR spectra is attributed to water in CD₃CN, as stated in the reference (*Organometallics* **29**, 2176–2179 (2010)). We have now annotated the peak on the NMR spectra provided in the Supplementary Information.

Reviewer #2 (Remarks to the Author):

Review of Kim et al – Nature Comm 2024 Iterative SuFEx approach for sequence-regulated oligosulfates..

Kim et al have provided a nice paper outlining the synthesis of sequence-regulated oligosulfates. Following (quoted) work of e.g. Niu (ACIE 2018) who pioneered the use of SuFEx towards sequence-defined oligomers, the authors noted that this had not been done for oligosulfates yet. The key of the SuFEx reaction as a click reaction is that the S-F bond does not react with almost anything else apart from O and N-centered nucleophiles, and the list of tolerated groups is in this paper extended with sulfur imidazylates. [Although this is THE key moiety of the entire paper, the authors do not give its structure, neither in the paper nor anywhere in the supporting information.. One is obliged to look at their previous Org Lett 2020 paper = ref 44 to be elucidated.]

The key issue is evaluate the novelty of this paper is 2-fold: is this surprising, and do the authors do something truly novel with it?

The answer to the first is: more yes than no, but not fully clear. The imidazole moiety is a relatively strong electron-donating group -> to replace it in a bi-functional molecule outfitted with both an S-F and an S-Im bond with an oxygen-centered nucleophile, any reaction at the S-Im site will be outcompeted by the F-Si bond-formation catalysis of the SuFEx variant chosen by the authors. In other words: the S-Im bond will be stable, and as the authors show it is basically orthogonal to the silyl-protected SuFEx variant. I might (and perhaps: would) have guessed the first, but that the authors were able to show that it is truly orthogonal and that it can be used to form even more complex structures in typical click yields, I would not have guessed -> with some consideration, I thus think they take the “Nat Comm-novelty hurdle” here.

And do they do something novel with this? This is a resounding yes: the structures have not been made before, and they display the variation that one would expect for a high-level paper! The structures shown in Figures 1-3 are simply beautiful. They will make many people in the field think: wow, I wish my group would have (been able to) do(ne) that! The authors do not only make ‘pretty pictures’, but the Supp Info provides the supporting data with convincing clarity -> really nice stuff!

After concluding the paper should be published in Nature Comm the question is: can it be/does it need to be improved? Yes, also here a resounding yes.

Response) We appreciate the reviewer's valuable comments on our manuscript. To address each of the comments, we have provided point-by-point responses below.

Apart from outlining to the broad readership of this journal what ‘Im’ is, I would suggest to improve

the manuscript as follows before acceptance:

Response) As per your suggestion, we have included the structure of -Im in the updated Fig. 1c in the main text.

Q1) Polysulfates have been shown to be superior materials for thin-film capacitors (Sharpless, Wu, Joule 2023). Pls add this ref., as it puts the usefulness of these materials in a wider context.

Response) We have revised the main text (highlighted in yellow on page 6) to emphasize the utilities of SuFEx-based polymers including polysulfates, and have incorporated new references (main text, Refs. 51–62) accordingly, including the *Joule* 2023 article. Please see also the response to Q2 of the Reviewer #1.

Q2) The paper presents many references to older reviews of SuFEx – given the rapid development of this field, likely any reviews prior to 2021 are becoming superfluous to introduce people to the field, as e.g. shown by the involvement of Sharpless himself in e.g. the *Nat* -series 2023 review. [= ref 20]

Response) We have included recent review articles in accordance with the reviewer's comment.

(Main text, Ref. 20) Zeng, D., Deng, W.-P. & Jiang, X. Advances in the construction of diverse SuFEx linkers. *Natl. Sci. Rev.* **10**, nwad123 (2023).

(Main text, Ref. 21) Moorhouse, A. D., Homer, J. A. & Moses, J. E. The certainty of a few good reactions. *Chem* **9**, 2063–2077 (2023).

Q3) The key feature is the stability of the S-Im group under SuFEx conditions. Yet, no clarity is provided on why that is indeed the case. The paper would thus be helped significantly by some quantum chemistry that calculated the transition states and driving forces of both reactions (for reaction at the S-F side and at the S-Im side), to provide insight in the chemoselectivity rather than just observations.

Response) We thank this reviewer for the valuable suggestion, as we fully agree that the contrasting stability of the S-Im group should be rationalized to explain the superb selectivity observed in the current SuFEx system. While the mechanism of base-catalysed activation of S(VI) compounds with silyl ethers is still in much debate, two reaction modes have been commonly proposed in this type of transformations: 1) reaction initiated by deprotection of silyl group with the aid of base as a catalyst (silyl ether activation), and 2) activation of sulfate compounds by base catalyst followed by reaction with silyl ethers (*ACS Catal.* **11**, 6578–6589 (2021)). As suggested by this reviewer, we

performed Density Functional Theory (DFT) calculations based on these two mechanistic scenarios to gain detailed insights about the superior reactivity of fluorosulfates over imidazolates, using simple substrates **SM_F**, **SM_Im**, and **SM_TBS** to save the costs of computational resources and minimize the potential errors that could be caused by complexity from many possible conformations (Fig. R1).

Fig. R1 | Energy profiles of the DBU-catalysed reaction of sulfates with silyl ethers; calculations were performed at the ω B97X-D/6-311+G**/SMD(acetonitrile)// ω B97X-D/6-31G** level of theory.

Addition of 1,8-diazabicyclo[5.4.0]undec-7-ene (**DBU**) to silyl ether **SM_TBS** was found to require a high activation barrier ($\Delta G^\ddagger = 41.0$ kcal/mol), suggesting that the deprotection of a silyl group by **DBU** is kinetically less feasible in the current system presumably due to the steric clashes as the major contributing factor (grey). **DBU** activation of sulfates was thus next considered, whereby both addition-elimination and direct substitution mechanisms might be possible (*Angew. Chem. Int. Ed.* **53**, 9430–9448 (2014)). While the transition states could not be located for the direct substitution pathway, DFT calculations revealed that the addition-elimination process is energetically feasible, which is in good agreement with the lowest unoccupied molecular orbitals (LUMO) on the sulfate part being mainly located at the rare site of the S=O bond (Fig. R2). The addition process would be the rate-determining step in this mechanism, hence, the overall reaction would be significantly influenced by electronics. Indeed, addition of **DBU** to the electronically more deficient fluorosulfate **SM_F** was calculated to traverse the corresponding transition state **TS1_F** with lower activation energy (deep blue; $\Delta G^\ddagger = 34.0$ kcal/mol) compared to the imidazolyl system stabilized through a

relatively electron-donating imidazole moiety (dark yellow; **TS1_Im**; $\Delta G^\ddagger = 40.9$ kcal/mol). Moreover, the succeeding elimination of fluoride in the fluorosulfate case would be thermodynamically driven by the formation of a low-energy fluorosilane species (**TBS-F**) accompanied by deprotection of **SM_TBS** to phenolate, whereas the analogous addition-elimination reaction of imidazylate **SM_Im** is overall an uphill process ($\Delta G = 13.4$ kcal/mol). The subsequent addition of phenolate to DBU-sulfonate intermediate **int2** and elimination of DBU to furnish the sulfate product **P** was also found to be energetically favorable.

Fig. R2 | LUMO of SM_F and SM_Im.

With the above computational results suggesting the involvement of base-catalysed sulfate activation mechanism, the superior selectivity of the current system toward fluorosulfates over imidazylates is likely originated from both the kinetic and thermodynamic influences. In order to comprehensively understand the underlying mechanism, our upcoming study will involve the combined approach of computational and experimental investigations to determine the influence of electronic and steric effects on this reaction. For the detailed computational information, please see also the Appendix (pages R15–R34) included in this response letter.

Q4) Polycondensation reactions are expected to give polymers with dispersity D ‘somewhat’ smaller than 2.0. This is what is also observed for all SuFEx-based polymers up to now (papers by Dong, Wu, Sharpless, Zuilhof, etc.) Yet, the current authors report D values of 1.03 and 1.06 for two of their three polymers. These numbers deserve some more attention, as also proven by their broad GPC traces, which are more in line with previous literature data, and less with mono-disperse polymers -> I would suggest the authors to reinvestigate and recalculate these, as there likely is no monodispersity observed for their polymers (in all honesty: this is also not expected, and nothing against the quality of the paper)

Response) We comprehensively reinvestigated the M_n and D values of **P-1**, **P-2**, and **P-3** (see also the updated GPC chromatogram, Fig. R3), using three-column analysis instead of the previous one-column analysis. The modifications made in the main text (page 5 and Fig. 5a) have been highlighted

in yellow. Additionally, detailed measurement conditions of the gel permeation chromatography (GPC) analysis have been provided in the Supplementary Information (pages 2, 32, and Fig. S1).

Fig. R3 | GPC chromatogram of P-1, P-2, and P-3 through three-column analysis.

With the aim of improving the accuracy of M_n and D values, we implemented a three-column system in place of the previous one-column system. The incorporation of additional GPC columns enables finer resolution in separating polymer molecules based on size. This refinement leads to a more precise determination of molecular weights as the separation becomes more reliable. The increased column count enhances separation efficiency, allowing for the detection of a broader spectrum of polymer sizes. Consequently, this advancement results in a more precise characterization of the molecular weight distribution within the sample. The three-column GPC analysis of **P-1**, **P-2**, and **P-3** gave M_n values of 19 kDa ($D = 1.5$), 24 kDa ($D = 1.7$), and 48 kDa ($D = 2.0$), respectively.

The detailed conditions of GPC analysis are described below. (See also the updated Supplementary Information, 1.1. Characterisation)

(One-column system) GPC analysis was performed using an Agilent 1200 infinity series equipped with a Waters Styragel HR-3 column for the GPC trace monitoring. The system was calibrated with polystyrene calibration kit from Agilent Technologies. DMF with 0.05 M LiBr was used as a mobile phase with an eluent flow rate of 1.0 mL/min and a column temperature of 40 °C.

(Three-column system) GPC analysis was conducted using a Jasco LC-4000 Series equipped with three polystyrene-gel columns in series (Shodex; KD-802, KD-803, and KD-804), along with guard column. This method was used for the M_n and D values. Reflex index detector (RI-4030) and UV detector (UV-4075) were employed. DMF with 0.05 M LiBr was used as a mobile phase with an eluent flow rate of 0.2 mL/min and a column temperature of 45 °C. Results were calibrated by Shodex polystyrene standard ($M_p = 1230\text{--}2400000$; $D = 1.02\text{--}1.08$).

Q5) The authors artificially divide the paper in a Results & Discussion section, but in truth their Results contain some discussion (as it should: there are no uninterpreted data), and their Discussion is ‘just’ a Conclusion section under a different name. I would suggest to rename them as such.

Response) We have renamed the section titles in the main text, as suggested by the reviewer.

Q6) It is already known that the SuFEx linkages are stable under neutral and acidic conditions, and break down under basic conditions. The authors also do this, and present their case, by using 0.30 mmol in 0.1 mL solutions (these conditions are only mentioned to those readers stubborn enough to continue searching the Supp Info to p S27). This is thus a 3 M (!) NaOH solution... -> almost any heterolytic bond will break down by heating overnight under these conditions.. The authors do also not compare their results with the more gentle breakdown of ref 36, which I think presented the first breakdown of SuFEx-based polymers, nor study the development of the macromolecular mass over time in neutral media at/near R.T. (which is what is most interesting in view of any applications). Here some more experimental work would surely improve the paper further.

Response) In accordance with the reviewer’s suggestion, we performed additional experiments to reveal the changes of polysulfate **P-2**. These experiments involved the monitoring GPC traces over time, as depicted in Fig. R4.

Table R1. The degradation tests using NaOH, NaI, or PPh₃

Entry	Reagent	Temperature	Result
1	NaOH (12 mg, 0.30 mmol)	rt	Minor change
2	NaI (45 mg, 0.30 mmol)	rt	No significant change
3	PPh ₃ (79 mg, 0.30 mmol)	rt	No significant change
4	NaOH (12 mg, 0.30 mmol)	80 °C	Full degradation

* Standard conditions: **P-2** (35 mg, 0.03 mmol), DMF (0.2 mL), 16 h.

The degradation tests were performed under various conditions, as outlined in Table R1 (entries 1–4). A volume of 6 μL of each mixture was taken and diluted in 1.0 mL of DMF/LiBr solution. Following this, filtration was performed using a syringe filter, and the samples were subjected to

analysed via GPC. The GPC traces have been incorporated into the Supplementary Information, Section 7.3.

Fig. R4 | GPC chromatograms of entries 1–4.

As shown in Fig. R4 (entry 3), we examined the degradation conditions involving PPh_3 . After 16 h, no significant changes were observed in the GPC chromatogram. This lack of observable degradation can be attributed to the fact that triphenylphosphine is known to facilitate the disulfide cleavage in the corresponding reference (*Angew. Chem. Int. Ed.* **61**, e202116158 (2022)), whereas our polymeric structures do not contain a disulfide moiety.

Finally, it was a joy to read the paper – it is clear, specific enough to the specialist yet (apart from the ‘missing-Im’ enticing enough to draw also non-specialists to this field -> great fit for Nature Comm, I think!

Response) Thank you again for your kind feedback.

Reviewer #3 (Remarks to the Author):

In this paper, Kim et al. reported an iterative method for the synthesis of oligosulfates based on a chain homologation approach, in which the fluorosulfate was regenerated by treating imidazylate intermediate with AgF, a methodology developed by part of the authors before. Polysulfate periodic copolymers are also synthesized from oligomeric bisfluorosulfates. Although several new oligosulfates and polysulfate periodic copolymers were made, it is more like short multiple-step synthesis with an imidazylate as the protecting group plus reliable SuFEx reactions. From the practical perspective, many other regular protecting groups, which could be deprotected in excellent yields before the re-installation of a fluorosulfate to one side of the molecule in one pot, might be more suitable for this task. Overall, I think the scientific contributions of this paper do not meet the high-quality requirements of Nat. Commun. Thus I don't support the publication of the paper in this Journal.

Response) The Niu group reported the seminal synthetic work accessing sequence-regulated structures based on oligosulfate- and triazole moieties through the combination use of SuFEx and CuAAC click reactions (*Angew. Chem. Int. Ed.* **57**, 16194–16199 (2018)). However, to date, the synthesis of discrete oligosulfates solely constructed by $-(OSO_2-O)-$ backbone scaffold remains unestablished. In this study, we present the first iterative synthesis of sequence-regulated oligosulfates and polymers, showcasing their application as adhesives.

Our group previously reported the conversion of aryl imidazylates to aryl fluorosulfates (*Org. Lett.* **22**, 5511–5516 (2020)). Yet, the reactivity difference between aryl imidazylate and aryl fluorosulfate towards silyl ether in the presence of DBU has not been elucidated. Notably, the chemoselective coupling of aryl fluorosulfate with silyl ether is the essence of this homologative iterative strategy, while preserving imidazylate moiety intact. We also conducted the DFT calculations (see also the response to Q3 by the Reviewer #2), revealing that the chemoselectivity largely stems from the lower energy barrier for the fluorosulfate compared to the imidazylate.

Fig. R5 | Comparison of three-step PG approach and two-step homologation approach. a Representative building blocks. **b** Three-step PC approach. **c** Two-step homologation approach.

The iterative assembly of synthetic building blocks, such as amino acids or monosaccharides, has successfully yielded sequence-regulated structures, such as oligopeptides or oligosaccharides. Similarly, the protecting group (PG) approach can be applied to the synthesis of complex oligosulfates. However, our homologation method offers distinct synthetic advantages compared to the PG approach. As illustrated in Fig. R5, aryl imidazolylate cannot be considered as a PG, but it acts as a transient group. While our homologation approach involves coupling and subsequent fluorination to regenerate fluorosulfate moiety, the conventional PG approach consists of three-step operations of coupling, deprotection, and final SO_2F installation. Consequently, as the chain length with a discrete sequence increases, the PG approach inevitably faces diminished step economy over repeated cycles.

We next evaluated synthetic efficacy and reliability between our homologation and conventional PG approaches. As shown in Fig. R6, the homologation approach afforded the $\text{MeO}\{-\mathbf{A}-\mathbf{A}-\mathbf{C}\}-\text{OSO}_2\text{F}$ sequenced target trimer **22** in four steps, achieving an overall yield of 76%. Conversely, the PG approach, both under acidic- and basic conditions, gave lower product yields, due to the lengthier synthesis and inefficient deacetylation for sequence cleavage. This result indicates that the homologation approach in this work demonstrates superior synthetic efficacy in terms of both step economy and reliability.

Fig. R6 | Sequence-regulated oligosulfates through PG approach. Reaction conditions: (i) fluorosulfate (1.0 equiv), building block (1.0 equiv), DBU (25 mol %), MeCN, 80 °C, 2 h; (ii) AcCl (15 mol %), MeOH (0.9 M), rt, 3 h; (iii) K₂CO₃ (2.5 equiv), MeOH, rt, 4 h; (iv) SuFEx-IT (1.3 equiv), E₃N (1.6 equiv), MeCN, 0 °C to rt, 16 h; (v) AcCl (15 mol %), MeOH (0.5 M), rt, 16 h.

Both acetyl protecting group and sulfate linkage exhibit acid- and base-lability. As depicted in Fig. R7, acidic cleavage conditions provide the deacetylation product **19''** in 50% yield, along with the remaining substrate in 38%. With a longer reaction time, an undesired product (**Side P**) was observed, likely due to the sulfate cleavage of substrate **19'** followed by the condensation between the cleaved one with the substrate **19'**. The formation of the side product is detrimental for the synthesis of sequence-regulated structures. The basic deacetylation step also yielded the undesired **Side P** in 6%. While PG approach encounters the issue of oligosulfate cleavage under deprotection conditions, the homologation approach employed in this work, performed under the neutral conditions, offers a

more practical and reliable synthetic route for an array of discrete oligomers. The above-discussion has been included in the Supplementary Information, Section 9.

Fig. R7 | Formation of side products from acidic and basic deprotection steps.

Appendix. Computational Details

All calculations were performed using the density functional theory (DFT) as implemented in the Gaussian 09 suite of programs. Geometry optimizations were performed using the ω B97X-D functional and 6-31G** basis set. Vibrational frequency calculations were carried out at the same level of theory as that used for geometry optimizations, wherein thermochemistry correction energy ($G - E$) was acquired. Transition states were realized by the presence of single imaginary frequency. Single point energies of optimized structures were calculated with the ω B97X-D functional and 6-311+G** basis set. Solvation effects were incorporated using the SMD model (solvent = acetonitrile) based on the gas-phase optimized geometries and carried out with at the same level as single-point calculations. Final solution phase Gibbs free energies were calculated as follows:

$$G_{\text{sol}} = E_{\text{sol}} + (G - E) \quad (1)$$

$$\Delta G(\text{sol}) = \Sigma G(\text{sol}) \text{ for products} - \Sigma G(\text{sol}) \text{ for reactants} \quad (2)$$

Summarized energy components of all DFT-optimized structures

Supplementary Table R2. Summarized energy components of DFT-optimized structures

DFT-optimized Structures	E(sol) (SCF/TZ) [eV] ωB97X-D/6-311+G**/ SMD(acetonitrile)	G–E (Thermochemistry correction energy) [eV] ωB97X-D/6-31G**	G(sol) [eV]
SM_F	-27063.855468	2.649138	-27061.206329
SM_Im	-30485.983183	4.166038	-30481.817145
SM_TBS	-22697.236770	6.786740	-22690.450030
PhO ⁻	-8353.033733	1.716985	-8351.316749
DBU	-12574.214882	5.826395	-12568.388487
TBS-DBU ⁺	-26917.900563	11.030847	-26906.869716
TBS-F	-17064.968616	4.455376	-17060.513240
TBS-Im	-20486.391572	5.954452	-20480.437120
P	-32696.720102	4.946243	-32691.773860
TS1_TBS	-35270.431226	13.371517	-35257.059709
int1_TBS	-35270.808206	13.414592	-35257.393614
TS1_Im	-43059.071908	10.640717	-43048.431191
int1_Im	-43059.715820	10.640527	-43049.075294
TS2_Im	-43059.759813	10.713562	-43049.046251
TS1_F	-39637.220476	9.099001	-39628.121475
int1_F	-39637.950143	9.173451	-39628.776692
TS2_F	-39637.778938	9.157941	-39628.620997
int2	-36917.474387	9.151710	-36908.322678
TS3	-45270.306720	11.493658	-45258.813061
int3	-45270.318677	11.485468	-45258.833209
TS4	-45269.862424	11.435399	-45258.427025

Appendix. Cartesian Coordinates of All Optimized Geometries

=====

SM_F

=====

Charge: 0, Spin: 1

Cartesian coordinates:

ATOM	X	Y	Z
C	-1.602036000	5.282654000	-0.374384000
C	-2.569248000	5.434657000	-1.354580000
C	-1.651165000	5.970555000	0.827447000
H	-2.493261000	4.875826000	-2.280083000
H	-0.874089000	5.820788000	1.567961000
C	-3.620693000	6.312209000	-1.114226000
C	-2.711576000	6.842693000	1.045993000
H	-4.385005000	6.444456000	-1.874311000
H	-2.762310000	7.391487000	1.981795000
C	-3.709677000	7.025304000	0.084519000
O	-0.497153000	4.441524000	-0.632779000
O	-0.703870000	2.809974000	1.251336000
S	-0.691860000	2.916960000	-0.177652000
O	-1.656998000	2.257478000	-1.006667000
F	0.742550000	2.488733000	-0.678235000
C	-4.871162000	7.949236000	0.347493000
H	-5.270574000	8.362059000	-0.582363000
H	-5.685064000	7.413474000	0.847983000
H	-4.579370000	8.782149000	0.992409000

=====

SM_Im

=====

Charge: 0, Spin: 1

Cartesian coordinates:

ATOM	X	Y	Z
C	-4.341545000	9.271779000	0.931993000
C	-3.368254000	9.939479000	0.186567000
C	-4.816649000	9.874473000	2.101976000
H	-2.987449000	9.488023000	-0.724719000
H	-5.574460000	9.369410000	2.694161000
C	-2.874343000	11.176986000	0.588784000
C	-4.338335000	11.109085000	2.521227000
H	-2.121535000	11.702656000	0.013074000
H	-4.704395000	11.584895000	3.423557000
C	-3.369378000	11.739345000	1.753605000
O	-2.835162000	12.963300000	2.202644000
S	-3.578508000	14.290382000	1.645669000
O	-4.904375000	14.364578000	2.201209000
O	-3.320425000	14.432131000	0.236298000
N	-2.618075000	15.401401000	2.445224000
C	-1.451686000	15.970797000	1.966200000
C	-2.773754000	15.831397000	3.758846000
C	-1.706022000	16.641449000	3.983135000
N	-0.893551000	16.722221000	2.866303000
H	-1.110122000	15.789161000	0.957926000
H	-1.470675000	17.185862000	4.885631000
H	-3.618964000	15.530004000	4.355094000
C	-4.882055000	7.934940000	0.492412000
H	-4.358627000	7.563195000	-0.391525000
H	-4.778072000	7.188294000	1.285681000
H	-5.946623000	8.006028000	0.247152000

=====

SM_TBS

=====

Charge: 0, Spin: 1

Cartesian coordinates:

ATOM	X	Y	Z
C	0.024824000	-0.006240000	0.000178000
C	0.411924000	-1.187267000	-0.634855000
C	-0.836163000	0.882629000	-0.645252000
H	1.089945000	-1.861460000	-0.121666000
H	-1.118444000	1.800316000	-0.139549000
C	-0.067594000	-1.476343000	-1.908637000
C	-1.310024000	0.584355000	-1.918915000
H	0.237387000	-2.396209000	-2.398220000
H	-1.979109000	1.279769000	-2.416482000
C	-0.930484000	-0.594933000	-2.555366000
O	0.509705000	0.293288000	1.239982000
Si	-0.270194000	-0.176392000	2.664254000
C	-0.380284000	-2.049018000	2.718379000
H	-0.905292000	-2.384741000	3.618778000
H	-0.931895000	-2.425291000	1.850770000
H	0.611262000	-2.511609000	2.710937000
C	0.823839000	0.519460000	4.040973000
C	-1.998708000	0.553912000	2.689899000
H	-2.572255000	0.211856000	1.822378000
H	-2.540459000	0.246404000	3.590449000
H	-1.974726000	1.647551000	2.664852000
C	0.910369000	2.049711000	3.910032000
H	1.561534000	2.460851000	4.692712000
H	1.322396000	2.345248000	2.940091000
H	-0.071691000	2.523177000	4.019889000
C	0.219013000	0.155908000	5.408211000
H	0.153243000	-0.928808000	5.551026000
H	0.844673000	0.554556000	6.217382000
H	-0.785455000	0.575576000	5.535703000
C	2.237524000	-0.077003000	3.931408000
H	2.885732000	0.338227000	4.714400000
H	2.230353000	-1.165584000	4.056429000
H	2.693065000	0.149707000	2.962396000
H	-1.301504000	-0.823637000	-3.549026000

=====

PhO⁻

=====

Charge: -1, Spin: 1

Cartesian coordinates:

ATOM	X	Y	Z
C	-0.002311000	0.003219000	0.004381000
C	-0.778431000	-0.925059000	0.799181000
C	0.778148000	0.925984000	0.801424000
H	-1.387347000	-1.645530000	0.254584000
H	1.384027000	1.650232000	0.258453000
C	-0.766651000	-0.920426000	2.183868000
C	0.773977000	0.911679000	2.186056000
H	-1.376948000	-1.649861000	2.720900000
H	1.387189000	1.637348000	2.724868000
C	0.005632000	-0.006925000	2.912906000
O	-0.005747000	0.007690000	-1.256081000

H 0.008651000 -0.010748000 4.000327000

=====
DBU
=====

Charge: 0, Spin: 1

Cartesian coordinates:

ATOM	X	Y	Z
N	-0.330790000	-0.684770000	-0.402311000
C	-0.337148000	0.698719000	-0.325339000
N	-1.313042000	1.432340000	0.070518000
C	-2.541045000	0.794252000	0.510745000
C	-2.736908000	-0.615003000	-0.042800000
C	-1.456745000	-1.410338000	0.166792000
C	0.899816000	-1.436833000	-0.597195000
C	1.953274000	-1.264224000	0.506950000
C	2.812260000	-0.009122000	0.337478000
C	2.037837000	1.310547000	0.349174000
C	0.934319000	1.418984000	-0.717813000
H	-2.547775000	0.765136000	1.610136000
H	-3.381927000	1.433844000	0.220091000
H	-2.950698000	-0.563670000	-1.116050000
H	-3.578629000	-1.119241000	0.441995000
H	-1.525869000	-2.385329000	-0.327916000
H	-1.301539000	-1.600321000	1.241413000
H	0.609572000	-2.489304000	-0.662080000
H	1.344786000	-1.185402000	-1.568618000
H	1.445822000	-1.253988000	1.480071000
H	2.610841000	-2.141561000	0.502630000
H	3.574930000	0.015558000	1.124158000
H	3.354725000	-0.089962000	-0.615200000
H	2.749602000	2.129518000	0.198072000
H	1.582654000	1.472751000	1.334027000
H	1.309798000	1.065229000	-1.686310000
H	0.643751000	2.463257000	-0.836761000

=====
TBS-DBU⁺
=====

Charge: 1, Spin: 1

Cartesian coordinates:

ATOM	X	Y	Z
Si	-0.560125000	3.558905000	-1.392917000
C	-1.546303000	3.568701000	-3.007793000
C	1.263205000	3.934022000	-1.629058000
H	1.867912000	3.602054000	-0.780493000
H	1.704293000	3.533630000	-2.545082000
H	1.358798000	5.023874000	-1.679572000
C	-1.204553000	4.808182000	-0.153126000
H	-2.293236000	4.862449000	-0.079425000
H	-0.801002000	4.633931000	0.849170000
H	-0.856345000	5.797144000	-0.468596000
C	-1.412357000	4.981602000	-3.611674000
H	-1.815800000	5.753388000	-2.947484000
H	-0.370991000	5.235707000	-3.837904000
H	-1.971874000	5.038995000	-4.552522000
C	-1.007095000	2.534969000	-4.010745000
H	0.045646000	2.707413000	-4.261386000

H	-1.112292000	1.509302000	-3.638131000
H	-1.572573000	2.594881000	-4.947814000
C	-3.031046000	3.267689000	-2.735916000
H	-3.174866000	2.256468000	-2.337142000
H	-3.480036000	3.982378000	-2.037729000
H	-3.602186000	3.328749000	-3.669449000
N	-0.505240000	-0.376661000	-0.167072000
C	-0.188369000	0.762713000	-0.783951000
N	-0.854788000	1.904094000	-0.592045000
C	-1.992820000	1.902756000	0.354439000
C	-2.702054000	0.564918000	0.349153000
C	-1.700341000	-0.519790000	0.675172000
C	0.355464000	-1.575904000	-0.241553000
C	1.742364000	-1.397372000	0.380892000
C	2.757311000	-0.701943000	-0.526156000
C	2.345891000	0.703047000	-0.961637000
C	0.997330000	0.739965000	-1.709448000
H	-1.622263000	2.145039000	1.356766000
H	-2.685905000	2.689515000	0.061373000
H	-3.147978000	0.384352000	-0.634399000
H	-3.507626000	0.563595000	1.085985000
H	-2.124552000	-1.507266000	0.478962000
H	-1.400987000	-0.485868000	1.729106000
H	-0.183701000	-2.353524000	0.300186000
H	0.428202000	-1.915538000	-1.280526000
H	1.635555000	-0.857608000	1.329720000
H	2.118707000	-2.394080000	0.630614000
H	3.722035000	-0.651109000	-0.013374000
H	2.914208000	-1.318530000	-1.420504000
H	3.109979000	1.113703000	-1.627267000
H	2.298810000	1.370151000	-0.092867000
H	0.911769000	-0.126460000	-2.372857000
H	0.949666000	1.610528000	-2.354585000

=====
TBS-F
=====

Charge: 0, Spin: 1

Cartesian coordinates:

ATOM	X	Y	Z
Si	-0.942347000	3.495847000	-0.390382000
C	-1.669310000	3.692996000	-2.125661000
C	0.907757000	3.775652000	-0.374710000
H	1.310805000	3.639811000	0.633445000
H	1.424250000	3.077169000	-1.039463000
H	1.152592000	4.793086000	-0.697282000
C	-1.793981000	4.606498000	0.851588000
H	-2.867069000	4.400764000	0.903776000
H	-1.375156000	4.460155000	1.851811000
H	-1.664394000	5.661181000	0.587047000
C	-1.451199000	5.137677000	-2.608842000
H	-1.945386000	5.866777000	-1.956760000
H	-0.387451000	5.396101000	-2.660903000
H	-1.867708000	5.266259000	-3.616318000
C	-0.973351000	2.721429000	-3.094473000
H	0.100747000	2.923649000	-3.170444000
H	-1.099431000	1.679945000	-2.781381000
H	-1.398069000	2.820633000	-4.102035000

C	-3.176992000	3.388271000	-2.095365000
H	-3.375586000	2.368138000	-1.751168000
H	-3.717485000	4.077846000	-1.437563000
H	-3.605360000	3.490461000	-3.101085000
F	-1.205185000	1.953519000	0.075336000

=====
TBS-Im
=====

Charge: 0, Spin: 1

Cartesian coordinates:

ATOM	X	Y	Z
N	-3.804374000	14.570838000	-0.239953000
C	-4.894246000	13.882232000	-0.710438000
C	-2.931600000	14.628620000	-1.315024000
C	-3.546929000	13.985011000	-2.350675000
N	-4.780834000	13.518929000	-1.963376000
H	-5.752843000	13.681695000	-0.083381000
H	-3.176409000	13.834693000	-3.354828000
H	-1.974196000	15.122188000	-1.245322000
Si	-3.497052000	15.113824000	1.437178000
C	-2.576306000	13.727523000	2.347455000
C	-5.172005000	15.460076000	2.203222000
H	-5.724143000	16.195365000	1.610515000
H	-5.786416000	14.558332000	2.281522000
H	-5.050982000	15.864919000	3.213107000
C	-2.472179000	16.678278000	1.319612000
H	-1.491441000	16.496905000	0.869997000
H	-2.987017000	17.429980000	0.714056000
H	-2.305935000	17.102889000	2.314845000
C	-2.291598000	14.169631000	3.793754000
H	-1.660761000	15.065046000	3.831368000
H	-3.213894000	14.381004000	4.346789000
H	-1.762200000	13.375090000	4.335079000
C	-3.440037000	12.454712000	2.360549000
H	-4.383506000	12.606259000	2.896750000
H	-3.676165000	12.115301000	1.346183000
H	-2.905262000	11.640421000	2.866019000
C	-1.247475000	13.424560000	1.633965000
H	-1.408298000	13.105661000	0.598559000
H	-0.581037000	14.294492000	1.623144000
H	-0.717470000	12.614835000	2.151787000

=====
P
=====

Charge: 0, Spin: 1

Cartesian coordinates:

ATOM	X	Y	Z
C	-1.165805000	5.380511000	-0.705217000
C	-2.166354000	5.537373000	-1.651699000
C	-1.226959000	5.994326000	0.536445000
H	-2.083342000	5.036866000	-2.609329000
H	-0.426094000	5.843218000	1.250806000
C	-3.257219000	6.340528000	-1.338659000
C	-2.326810000	6.792821000	0.829198000
H	-4.046002000	6.474132000	-2.073358000
H	-2.384794000	7.281862000	1.797365000

C	-3.355953000	6.976845000	-0.098336000
O	-0.028286000	4.623162000	-1.037056000
O	-1.130138000	2.405826000	-1.433467000
S	-0.115524000	3.062291000	-0.646475000
O	-0.089698000	2.901910000	0.786499000
C	1.783988000	1.441389000	-1.153034000
C	2.550864000	1.084096000	-0.053634000
C	1.479501000	0.548214000	-2.170079000
H	2.752986000	1.816562000	0.718998000
H	0.868512000	0.873549000	-3.003850000
C	3.035416000	-0.218148000	0.021357000
C	1.970662000	-0.750640000	-2.080070000
H	3.637834000	-0.517137000	0.872648000
H	1.744110000	-1.464256000	-2.865200000
C	2.747240000	-1.133093000	-0.988757000
O	1.345392000	2.771589000	-1.265301000
H	3.128089000	-2.147088000	-0.924761000
C	-4.558944000	7.818846000	0.242937000
H	-4.998299000	8.267482000	-0.651834000
H	-5.333962000	7.210608000	0.722062000
H	-4.298242000	8.624433000	0.934437000

=====
TS1_TBS
=====

Charge: 0, Spin: 1

Imaginary frequency: $-108.5865 \text{ cm}^{-1}$

Cartesian coordinates:

ATOM	X	Y	Z
C	2.737188000	3.083743000	2.033724000
C	1.778952000	3.096332000	1.008257000
C	3.999755000	2.530503000	1.757432000
O	2.557530000	3.549138000	3.277298000
H	0.793505000	3.510331000	1.183989000
C	2.087362000	2.569200000	-0.243925000
H	4.737398000	2.532824000	2.555263000
C	4.293603000	2.006726000	0.507069000
H	1.330220000	2.589998000	-1.022719000
C	3.337577000	2.021030000	-0.507960000
H	5.277910000	1.585770000	0.322006000
H	3.565314000	1.612551000	-1.486950000
Si	1.189407000	4.391789000	4.032706000
C	0.347498000	5.191634000	5.635431000
C	-0.118724000	3.051951000	3.644459000
C	1.276204000	5.906644000	2.892627000
N	3.127130000	4.274183000	5.524189000
C	-1.097645000	5.545060000	5.208077000
C	0.235969000	4.285887000	6.869786000
C	1.029138000	6.491612000	6.090577000
H	0.307408000	2.203038000	3.105050000
H	-0.588211000	2.668930000	4.556296000
H	-0.926282000	3.464194000	3.028158000
H	1.356428000	5.618423000	1.841882000
H	0.442661000	6.604502000	2.998875000
H	2.193664000	6.457848000	3.131354000
H	-1.137989000	6.190416000	4.325487000
H	-1.698695000	4.655699000	4.999401000
H	-1.596612000	6.086160000	6.023169000

H	-0.251556000	3.328777000	6.650553000
H	1.217230000	4.086747000	7.307268000
H	-0.372033000	4.779769000	7.640433000
H	1.995258000	6.285891000	6.558071000
H	1.180729000	7.200181000	5.269508000
H	0.409115000	6.997571000	6.843615000
C	3.579108000	3.164148000	6.013905000
N	4.874960000	2.953365000	6.404968000
C	5.885299000	3.996502000	6.256684000
C	5.367735000	1.640183000	6.810184000
C	2.631921000	1.991077000	6.122611000
C	4.089437000	5.315055000	5.195086000
C	5.234799000	5.367229000	6.193507000
H	4.468658000	5.117794000	4.182739000
H	3.570713000	6.275809000	5.158054000
H	4.845866000	5.639817000	7.180752000
H	5.976810000	6.117485000	5.906604000
H	6.567397000	3.938160000	7.111974000
H	6.481284000	3.815423000	5.350395000
C	5.386776000	0.587953000	5.695489000
H	6.384792000	1.798596000	7.177116000
H	4.790912000	1.266103000	7.665532000
C	4.031943000	-0.084876000	5.476890000
H	5.732420000	1.063602000	4.769095000
H	6.126589000	-0.177819000	5.955648000
C	2.905634000	0.874563000	5.097390000
H	4.128662000	-0.847404000	4.696545000
H	3.756526000	-0.616937000	6.398648000
H	1.981488000	0.298833000	4.980700000
H	3.105653000	1.335691000	4.126281000
H	2.635171000	1.583824000	7.141626000
H	1.632142000	2.382225000	5.948889000

=====
int1_TBS
=====

Charge: 0, Spin: 1

Cartesian coordinates:

ATOM	X	Y	Z
C	2.032297000	1.066097000	0.161445000
C	3.103799000	0.169032000	0.357058000
C	2.354060000	2.404523000	-0.132812000
O	0.801646000	0.584846000	0.269400000
H	2.860281000	-0.865010000	0.590734000
C	4.423712000	0.585360000	0.259469000
H	1.561105000	3.126928000	-0.285388000
C	3.681582000	2.810793000	-0.228181000
H	5.223769000	-0.133631000	0.416125000
C	4.728212000	1.913640000	-0.035816000
H	3.897770000	3.851030000	-0.457222000
H	5.759642000	2.241604000	-0.112954000
Si	-0.951188000	1.336990000	-0.067795000
C	-2.912381000	1.621358000	-0.349433000
C	-0.392539000	1.782462000	-1.845613000
C	-0.672028000	2.657660000	1.278671000
N	-1.310429000	-0.458126000	0.635124000
C	-3.130513000	3.075066000	-0.811942000

C	-3.514135000	0.688166000	-1.414999000
C	-3.758364000	1.408036000	0.918937000
H	-0.189282000	2.859916000	-1.863485000
H	0.532515000	1.283613000	-2.141228000
H	-1.152329000	1.604728000	-2.610985000
H	-1.548215000	2.848216000	1.903771000
H	0.179248000	2.421441000	1.920743000
H	-0.442278000	3.605838000	0.778181000
H	-4.199410000	3.273500000	-0.978868000
H	-2.783133000	3.798130000	-0.065629000
H	-2.613378000	3.293397000	-1.752078000
H	-3.034110000	0.790344000	-2.394635000
H	-3.446856000	-0.363536000	-1.111706000
H	-4.581130000	0.908814000	-1.564574000
H	-3.425836000	2.017814000	1.765691000
H	-4.807797000	1.676783000	0.729146000
H	-3.756587000	0.359551000	1.235393000
C	-0.930440000	-1.547985000	0.015141000
N	-0.609121000	-2.680620000	0.673257000
C	-0.246032000	-2.682283000	2.095928000
C	-0.322087000	-3.933361000	-0.026214000
C	-0.826870000	-1.550386000	-1.487161000
C	-1.398691000	-0.522797000	2.096672000
C	-0.195582000	-1.271557000	2.649022000
H	-1.454060000	0.492205000	2.488068000
H	-2.330181000	-1.032500000	2.376469000
H	-0.210773000	-1.300744000	3.741134000
H	0.707071000	-0.754475000	2.313597000
H	-0.976901000	-3.292738000	2.641547000
H	0.730447000	-3.171053000	2.189413000
C	1.042692000	-3.952989000	-0.721017000
H	-0.373730000	-4.720380000	0.731051000
H	-1.121178000	-4.146227000	-0.743601000
C	1.045777000	-3.228363000	-2.068037000
H	1.785770000	-3.506756000	-0.048042000
H	1.339579000	-4.996615000	-0.873362000
C	0.605494000	-1.765324000	-2.007189000
H	2.051167000	-3.279692000	-2.498574000
H	0.384915000	-3.772329000	-2.757647000
H	0.655243000	-1.344113000	-3.016093000
H	1.288923000	-1.175086000	-1.389412000
H	-1.504584000	-2.312748000	-1.892116000
H	-1.208191000	-0.597396000	-1.845552000

=====
TS1_Im
=====

Charge: 0, Spin: 1

Imaginary frequency: $-145.8407 \text{ cm}^{-1}$

Cartesian coordinates:

ATOM	X	Y	Z
C	4.116956000	1.293445000	-2.652360000
C	5.127795000	0.836525000	-3.491240000
C	2.780839000	1.174483000	-3.028767000
H	6.159480000	0.918698000	-3.167608000
H	2.009202000	1.524946000	-2.352120000
C	4.798933000	0.275570000	-4.719248000
C	2.467042000	0.606123000	-4.256459000

H	5.590261000	-0.080723000	-5.373474000
H	1.424679000	0.511983000	-4.549353000
C	3.466863000	0.151874000	-5.122209000
O	4.448422000	1.940505000	-1.481117000
O	2.863569000	1.287152000	0.345452000
S	4.237511000	0.983799000	-0.022928000
O	4.696769000	-0.292251000	-0.565717000
C	3.114342000	-0.433851000	-6.466017000
H	2.982041000	0.354926000	-7.215324000
H	3.899478000	-1.103356000	-6.827516000
H	2.180522000	-1.001128000	-6.419581000
N	6.933609000	4.494990000	0.966876000
C	6.410230000	3.356586000	0.428058000
N	5.159741000	3.040168000	0.509904000
C	4.211830000	3.974562000	1.083518000
C	4.847825000	4.713259000	2.252945000
C	6.109100000	5.404682000	1.760907000
C	8.284593000	4.953910000	0.655744000
C	8.507780000	5.313417000	-0.817327000
C	8.794389000	4.103953000	-1.708401000
C	7.692199000	3.044562000	-1.732612000
C	7.337161000	2.457823000	-0.354478000
H	3.895521000	4.678018000	0.301917000
H	3.319594000	3.422534000	1.384635000
H	5.096834000	3.992474000	3.038894000
H	4.163107000	5.454392000	2.673665000
H	6.706799000	5.749098000	2.611670000
H	5.852437000	6.288198000	1.159179000
H	8.452099000	5.838884000	1.273859000
H	9.023865000	4.210371000	0.979795000
H	7.628785000	5.860252000	-1.180513000
H	9.357671000	6.002465000	-0.877905000
H	8.982879000	4.446949000	-2.731393000
H	9.725979000	3.634074000	-1.363638000
H	8.022939000	2.220809000	-2.374078000
H	6.776463000	3.440293000	-2.186195000
H	8.242984000	2.245365000	0.223806000
H	6.825937000	1.510197000	-0.508357000
N	4.942022000	0.733166000	1.586918000
C	4.486444000	1.269822000	2.759047000
H	3.505988000	1.713777000	2.835447000
N	5.346537000	1.112264000	3.733570000
C	6.422864000	0.465322000	3.163010000
C	6.197455000	0.222302000	1.838805000
H	6.761067000	-0.293052000	1.078994000
H	7.284753000	0.184265000	3.751354000

=====
int1_Im
=====

Charge: 0, Spin: 1

Cartesian coordinates:

ATOM	X	Y	Z
C	-2.207886000	0.002297000	0.121199000
C	-2.936696000	-0.989155000	-0.543186000
C	-2.904272000	1.041282000	0.748928000
H	-2.424515000	-1.794851000	-1.052808000
H	-2.334062000	1.805391000	1.269732000

C	-4.325605000	-0.925738000	-0.563599000
C	-4.291722000	1.093858000	0.705907000
H	-4.874113000	-1.706666000	-1.085160000
H	-4.807415000	1.915916000	1.197081000
C	-5.033011000	0.107507000	0.053678000
O	-0.855940000	0.076278000	0.183666000
O	-0.369942000	-2.135014000	1.007395000
S	0.217365000	-1.392514000	-0.104502000
O	-0.068812000	-1.464044000	-1.537222000
C	-6.540991000	0.137939000	0.040529000
H	-6.920593000	1.156887000	0.160988000
H	-6.939588000	-0.259825000	-0.897598000
H	-6.958786000	-0.466050000	0.854399000
N	2.080400000	2.095923000	0.542446000
C	1.525416000	1.089413000	-0.142336000
N	1.440154000	-0.115171000	0.405303000
C	1.971459000	-0.350347000	1.749538000
C	1.613829000	0.847028000	2.612915000
C	2.262795000	2.070785000	1.999896000
C	2.445154000	3.367632000	-0.095103000
C	1.248097000	4.265704000	-0.411780000
C	0.515418000	3.859970000	-1.690492000
C	-0.001638000	2.421806000	-1.693026000
C	1.085321000	1.339805000	-1.559161000
H	1.526208000	-1.265808000	2.134251000
H	3.053624000	-0.502134000	1.675465000
H	1.967715000	0.713647000	3.637292000
H	0.524799000	0.947735000	2.629234000
H	3.338383000	2.089733000	2.212716000
H	1.825422000	2.992099000	2.398850000
H	3.121165000	3.865760000	0.603833000
H	3.028385000	3.167189000	-0.999249000
H	0.562161000	4.253073000	0.444498000
H	1.611183000	5.294204000	-0.512003000
H	-0.325236000	4.542103000	-1.852958000
H	1.195972000	3.999485000	-2.541821000
H	-0.523370000	2.237330000	-2.636104000
H	-0.741528000	2.268410000	-0.902546000
H	1.977710000	1.615739000	-2.137094000
H	0.717642000	0.404898000	-1.973302000
N	1.689589000	-2.468820000	-0.204136000
C	2.703263000	-2.349091000	-1.106168000
H	2.669130000	-1.626077000	-1.909270000
N	3.675031000	-3.206036000	-0.891198000
C	3.266039000	-3.926550000	0.207170000
C	2.045689000	-3.490470000	0.647665000
H	1.394909000	-3.806501000	1.445896000
H	3.869386000	-4.727370000	0.611718000

=====
TS2_Im
=====

Charge: 0, Spin: 1

Imaginary frequency: -76.4096 cm⁻¹

Cartesian coordinates:

ATOM	X	Y	Z
C	4.080415000	1.388742000	-2.945501000
C	5.052077000	0.875496000	-3.792892000

C	2.729024000	1.183943000	-3.192873000
H	6.099607000	1.037021000	-3.563785000
H	1.992777000	1.584543000	-2.505689000
C	4.659492000	0.151645000	-4.912962000
C	2.354939000	0.452942000	-4.314473000
H	5.417105000	-0.254926000	-5.577039000
H	1.299641000	0.284040000	-4.509655000
C	3.308680000	-0.069875000	-5.192285000
O	4.477948000	2.167192000	-1.858545000
O	3.137985000	1.271464000	0.018922000
S	4.515417000	1.387872000	-0.380049000
O	5.430397000	0.295320000	-0.594339000
C	2.886902000	-0.833712000	-6.421750000
H	2.672824000	-0.150895000	-7.251500000
H	3.670531000	-1.520178000	-6.752698000
H	1.981616000	-1.417619000	-6.235009000
N	6.912716000	4.273391000	0.888862000
C	6.459258000	3.214749000	0.221810000
N	5.175475000	2.863344000	0.309828000
C	4.221006000	3.632971000	1.127625000
C	4.924545000	4.198918000	2.342874000
C	6.132835000	4.993509000	1.904952000
C	8.245119000	4.838681000	0.622226000
C	8.396001000	5.435459000	-0.779770000
C	8.689999000	4.407997000	-1.873702000
C	7.636768000	3.309589000	-2.010959000
C	7.405792000	2.510747000	-0.712712000
H	3.801514000	4.416165000	0.485636000
H	3.428686000	2.953810000	1.429787000
H	5.211313000	3.386115000	3.015724000
H	4.236419000	4.853372000	2.882876000
H	6.796906000	5.163234000	2.756631000
H	5.855722000	5.969690000	1.486731000
H	8.381037000	5.624656000	1.365921000
H	9.016164000	4.087102000	0.821720000
H	7.486663000	6.001697000	-1.016165000
H	9.216819000	6.159503000	-0.746276000
H	8.801877000	4.925732000	-2.831723000
H	9.659756000	3.939463000	-1.659228000
H	7.963252000	2.603740000	-2.780533000
H	6.680402000	3.717940000	-2.354944000
H	8.349191000	2.350087000	-0.183510000
H	7.017634000	1.524556000	-0.941428000
N	5.110501000	0.897224000	2.063493000
C	4.472331000	1.136633000	3.225030000
H	3.388477000	1.135211000	3.289108000
N	5.260446000	1.386219000	4.279453000
C	6.517592000	1.285931000	3.751210000
C	6.432664000	0.988157000	2.402561000
H	7.219988000	0.782123000	1.684346000
H	7.401935000	1.408140000	4.366699000

=====
TS1_F
=====

Charge: 0, Spin: 1

Imaginary frequency: -148.0251 cm⁻¹

Cartesian coordinates:

ATOM	X	Y	Z
C	4.129184000	1.395288000	-2.662572000
C	4.813482000	0.723400000	-3.671850000
C	2.743477000	1.481807000	-2.683966000
H	5.896028000	0.666070000	-3.632083000
H	2.233374000	2.009504000	-1.885232000
C	4.095238000	0.137424000	-4.704885000
C	2.037931000	0.896436000	-3.729773000
H	4.628014000	-0.387157000	-5.493627000
H	0.953804000	0.969007000	-3.749047000
C	2.699029000	0.212466000	-4.751756000
O	4.825777000	2.074954000	-1.681450000
O	4.652788000	-0.024198000	-0.491252000
S	5.587944000	1.089271000	-0.476185000
O	6.962677000	0.927991000	-0.902566000
F	5.685840000	1.166158000	1.181722000
C	1.931138000	-0.454613000	-5.864750000
H	1.806004000	-1.524743000	-5.665345000
H	0.933872000	-0.021138000	-5.977909000
H	2.452829000	-0.359169000	-6.821475000
N	5.296068000	5.308517000	1.016965000
C	5.048498000	4.008316000	0.671418000
N	5.898865000	3.257947000	0.056231000
C	7.197812000	3.742163000	-0.369439000
C	7.206640000	5.258727000	-0.496476000
C	6.616545000	5.873311000	0.762452000
C	4.424587000	6.043710000	1.930033000
C	4.317473000	5.447770000	3.339617000
C	3.311257000	4.299860000	3.448092000
C	3.612642000	3.094053000	2.557569000
C	3.709633000	3.427553000	1.058524000
H	7.954599000	3.400389000	0.346755000
H	7.431581000	3.264706000	-1.322687000
H	6.604868000	5.559091000	-1.360431000
H	8.224943000	5.624795000	-0.651949000
H	6.506989000	6.955526000	0.644110000
H	7.281065000	5.699459000	1.621758000
H	4.837408000	7.053312000	1.991803000
H	3.424212000	6.150067000	1.492350000
H	5.314040000	5.117260000	3.658269000
H	4.014570000	6.246140000	4.026550000
H	3.254750000	3.969384000	4.490813000
H	2.315378000	4.688878000	3.193419000
H	2.819067000	2.352215000	2.688836000
H	4.542053000	2.605998000	2.866618000
H	2.914311000	4.126088000	0.773541000
H	3.551100000	2.528260000	0.466676000

=====
int1_F
=====

Charge: 0, Spin: 1

Cartesian coordinates:

ATOM	X	Y	Z
C	4.087723000	1.534542000	-2.833414000
C	4.992991000	0.832636000	-3.629469000
C	2.737019000	1.551024000	-3.181837000
H	6.036044000	0.800489000	-3.335155000

H	2.041481000	2.079164000	-2.539409000
C	4.547497000	0.168791000	-4.766982000
C	2.305323000	0.885636000	-4.322960000
H	5.261115000	-0.378964000	-5.377885000
H	1.249731000	0.903399000	-4.583750000
C	3.200682000	0.186792000	-5.136868000
O	4.516913000	2.246432000	-1.750920000
O	3.224178000	1.092282000	-0.070758000
S	4.657997000	1.339490000	-0.184258000
O	5.715735000	0.439336000	-0.613002000
F	4.926380000	1.079071000	1.490158000
C	2.729555000	-0.507944000	-6.390252000
H	2.745880000	0.171394000	-7.250638000
H	3.366902000	-1.362319000	-6.635622000
H	1.704106000	-0.873097000	-6.281618000
N	5.203689000	5.152535000	1.034463000
C	4.717180000	3.900446000	0.953545000
N	5.348798000	2.987992000	0.251616000
C	6.631155000	3.275973000	-0.388779000
C	6.570185000	4.678308000	-0.961726000
C	6.286871000	5.643212000	0.174268000
C	4.727189000	6.097593000	2.052582000
C	4.957097000	5.633252000	3.494309000
C	3.903368000	4.655008000	4.018712000
C	3.785788000	3.353801000	3.226694000
C	3.475747000	3.563155000	1.732265000
H	7.418612000	3.179621000	0.367786000
H	6.791399000	2.524461000	-1.156930000
H	5.774919000	4.704323000	-1.711732000
H	7.512344000	4.946312000	-1.444990000
H	5.982482000	6.620525000	-0.214062000
H	7.183923000	5.792506000	0.788756000
H	5.283158000	7.020799000	1.877714000
H	3.669953000	6.334573000	1.884407000
H	5.959568000	5.192753000	3.561817000
H	4.959976000	6.523022000	4.133257000
H	4.124689000	4.422272000	5.065517000
H	2.928870000	5.162478000	4.014763000
H	2.978562000	2.749861000	3.649630000
H	4.692891000	2.748408000	3.309441000
H	2.749104000	4.371142000	1.599552000
H	3.034620000	2.659770000	1.319181000

=====
TS2_F
=====

Charge: 0, Spin: 1

Imaginary frequency: -189.0461 cm⁻¹

Cartesian coordinates:

ATOM	X	Y	Z
C	3.801777000	1.550756000	-3.057885000
C	4.789182000	1.137404000	-3.946895000
C	2.545569000	0.951231000	-3.078296000
H	5.767846000	1.601732000	-3.898846000
H	1.797638000	1.275169000	-2.364055000
C	4.511008000	0.131072000	-4.864505000
C	2.283319000	-0.056947000	-3.999083000
H	5.286487000	-0.190558000	-5.555158000

H	1.302575000	-0.525809000	-4.009605000
C	3.255681000	-0.480545000	-4.908410000
O	4.055140000	2.593508000	-2.195851000
O	3.260935000	1.716263000	-0.071105000
S	4.538688000	2.165789000	-0.596886000
O	5.700690000	1.352578000	-0.908598000
F	5.193486000	2.344606000	1.231067000
C	2.950431000	-1.551384000	-5.925887000
H	2.538598000	-1.117401000	-6.844366000
H	3.850999000	-2.107625000	-6.201060000
H	2.216152000	-2.265979000	-5.543335000
N	5.613831000	5.149775000	1.370539000
C	4.702326000	4.370651000	0.813202000
N	4.886799000	3.902550000	-0.455034000
C	6.006429000	4.308975000	-1.324379000
C	6.782844000	5.465549000	-0.717475000
C	6.948594000	5.197026000	0.768022000
C	5.404179000	5.768276000	2.684817000
C	5.160827000	4.763201000	3.816423000
C	3.691420000	4.364099000	3.952564000
C	3.144257000	3.599660000	2.749530000
C	3.309454000	4.304423000	1.398275000
H	6.670226000	3.449827000	-1.453413000
H	5.586712000	4.563831000	-2.298097000
H	6.257373000	6.416335000	-0.853131000
H	7.754578000	5.546454000	-1.210845000
H	7.517770000	5.986540000	1.259237000
H	7.446069000	4.236171000	0.943916000
H	6.301740000	6.354330000	2.886240000
H	4.571294000	6.479120000	2.619164000
H	5.781603000	3.877767000	3.640985000
H	5.499114000	5.223650000	4.750773000
H	3.567205000	3.748447000	4.849360000
H	3.097807000	5.274890000	4.118108000
H	2.073215000	3.428248000	2.894792000
H	3.628226000	2.626915000	2.655512000
H	2.964771000	5.347686000	1.454690000
H	2.684525000	3.804002000	0.662393000

=====
int2
=====

Charge: 1, Spin: 1

Cartesian coordinates:

ATOM	X	Y	Z
C	3.878713000	1.572910000	-2.909939000
C	4.550600000	1.295311000	-4.087913000
C	2.652003000	1.013098000	-2.596046000
H	5.513440000	1.749780000	-4.290325000
H	2.161670000	1.246813000	-1.657811000
C	3.955428000	0.419991000	-4.988942000
C	2.078192000	0.139178000	-3.512894000
H	4.467217000	0.188179000	-5.917786000
H	1.118021000	-0.312504000	-3.283943000
C	2.715325000	-0.168136000	-4.719034000
O	4.461577000	2.512103000	-2.005282000
O	4.518033000	1.212024000	0.136389000
S	5.348768000	1.873503000	-0.833971000

O	6.532567000	1.257334000	-1.362285000
C	2.069803000	-1.091957000	-5.718019000
H	1.470884000	-0.521572000	-6.435821000
H	2.818529000	-1.651392000	-6.283772000
H	1.406105000	-1.808097000	-5.228174000
N	5.618488000	5.171034000	1.291810000
C	5.100074000	4.159104000	0.619535000
N	5.886488000	3.383488000	-0.164927000
C	7.325001000	3.667735000	-0.390225000
C	7.577638000	5.143909000	-0.177486000
C	7.034597000	5.558231000	1.173522000
C	4.821918000	5.940016000	2.279182000
C	4.345898000	5.111630000	3.475063000
C	3.085585000	4.285909000	3.214719000
C	3.223801000	3.264259000	2.088346000
C	3.622136000	3.899180000	0.738643000
H	7.917110000	3.053918000	0.293851000
H	7.566241000	3.366792000	-1.408206000
H	7.100488000	5.729459000	-0.969103000
H	8.651304000	5.336046000	-0.220825000
H	7.085131000	6.641708000	1.291753000
H	7.595323000	5.099839000	1.995604000
H	5.483020000	6.734287000	2.625199000
H	3.984574000	6.427320000	1.768976000
H	5.169913000	4.470499000	3.810690000
H	4.146565000	5.813002000	4.290885000
H	2.803244000	3.767734000	4.135331000
H	2.259397000	4.969049000	2.978994000
H	2.267481000	2.757665000	1.938403000
H	3.944059000	2.482863000	2.348439000
H	3.116980000	4.862686000	0.613229000
H	3.293047000	3.278837000	-0.088283000

=====
TS3
=====

Charge: 0, Spin: 1

Imaginary frequency: -37.8769 cm^{-1}

Cartesian coordinates:

ATOM	X	Y	Z
C	4.385289000	2.042159000	-2.210762000
C	4.117664000	0.709388000	-2.570292000
C	4.198929000	3.041310000	-3.185149000
H	4.256976000	-0.078280000	-1.840439000
H	4.390259000	4.073716000	-2.901844000
C	3.681865000	0.406509000	-3.855109000
C	3.774068000	2.722670000	-4.468780000
H	3.477903000	-0.630288000	-4.109109000
H	3.641964000	3.515528000	-5.200498000
C	3.509340000	1.400188000	-4.816763000
O	4.835494000	2.422316000	-1.016826000
O	3.335199000	1.162846000	0.576548000
S	4.779781000	1.272350000	0.677595000
O	5.764528000	0.359402000	0.116152000
N	6.497278000	4.720685000	0.299869000
C	6.410946000	3.423792000	0.579848000
N	5.335274000	2.913602000	1.191226000
C	4.162377000	3.722140000	1.547672000

C	4.563003000	5.174309000	1.700964000
C	5.330019000	5.590379000	0.459684000
C	7.633536000	5.260476000	-0.460313000
C	7.793949000	4.646989000	-1.856972000
C	8.610048000	3.353747000	-1.861245000
C	7.966257000	2.208332000	-1.083496000
C	7.625126000	2.551080000	0.375700000
H	3.414831000	3.612740000	0.758257000
H	3.761965000	3.306185000	2.474066000
H	5.188047000	5.320734000	2.587733000
H	3.666434000	5.787833000	1.815055000
H	5.688394000	6.617915000	0.546720000
H	4.701831000	5.510572000	-0.435779000
H	7.451010000	6.333014000	-0.540938000
H	8.554794000	5.142885000	0.122351000
H	6.800153000	4.468955000	-2.284497000
H	8.292970000	5.387758000	-2.490798000
H	8.767509000	3.033969000	-2.895998000
H	9.605458000	3.569038000	-1.446546000
H	8.653392000	1.357620000	-1.061859000
H	7.053023000	1.870863000	-1.574989000
H	8.464494000	3.077401000	0.849185000
H	7.469008000	1.635632000	0.940181000
C	4.563797000	-0.334036000	2.757594000
C	3.270823000	-0.501622000	3.243810000
C	5.453367000	-1.403539000	2.745444000
H	2.589798000	0.341653000	3.231543000
H	6.452374000	-1.253487000	2.352650000
C	2.875400000	-1.744186000	3.724589000
C	5.043075000	-2.641056000	3.229010000
H	1.865391000	-1.871287000	4.105769000
H	5.741218000	-3.474150000	3.220632000
C	3.751246000	-2.833077000	3.724267000
O	4.985149000	0.914314000	2.346207000
C	3.302034000	-4.186280000	4.215991000
H	2.574315000	-4.094534000	5.027383000
H	2.825493000	-4.757008000	3.410922000
H	4.145465000	-4.777691000	4.583083000
H	3.172343000	1.148551000	-5.817182000

=====
int3
=====

Charge: 0, Spin: 1

Cartesian coordinates:

ATOM	X	Y	Z
C	3.996825000	1.452374000	-2.665375000
C	4.561936000	1.791846000	-3.896795000
C	3.020813000	0.451432000	-2.625224000
H	5.315834000	2.571284000	-3.930973000
H	2.557970000	0.178883000	-1.685617000
C	4.160443000	1.148180000	-5.062014000
C	2.642832000	-0.189426000	-3.798420000
H	4.614898000	1.433218000	-6.008121000
H	1.888153000	-0.970643000	-3.744740000
C	3.196127000	0.141371000	-5.037554000
O	4.428139000	2.165612000	-1.586632000
O	2.879675000	1.645654000	0.214681000

S	4.319741000	1.467523000	0.056906000
O	5.035739000	0.236004000	-0.276665000
C	2.745209000	-0.549924000	-6.300054000
H	1.777205000	-0.165239000	-6.641503000
H	3.462560000	-0.402625000	-7.112564000
H	2.630506000	-1.627299000	-6.145421000
N	7.157137000	4.221613000	0.838424000
C	6.520114000	3.192203000	0.252198000
N	5.225317000	3.036468000	0.408750000
C	4.428231000	4.004804000	1.158156000
C	5.231934000	4.458624000	2.361158000
C	6.530890000	5.064192000	1.864554000
C	8.522973000	4.603664000	0.456110000
C	8.668595000	5.048804000	-1.001839000
C	8.776738000	3.897538000	-2.003627000
C	7.578816000	2.950254000	-2.009441000
C	7.305323000	2.280349000	-0.648733000
H	4.191432000	4.837639000	0.486027000
H	3.498564000	3.518600000	1.444857000
H	5.418766000	3.584379000	2.990333000
H	4.680819000	5.199595000	2.944543000
H	7.246559000	5.168659000	2.686215000
H	6.359127000	6.062377000	1.441871000
H	8.789935000	5.432983000	1.113896000
H	9.219840000	3.788697000	0.684668000
H	7.821418000	5.698815000	-1.253414000
H	9.571321000	5.665282000	-1.072281000
H	8.915584000	4.311531000	-3.007768000
H	9.685111000	3.322492000	-1.777775000
H	7.746827000	2.157435000	-2.743398000
H	6.668624000	3.470437000	-2.324288000
H	8.243761000	2.010057000	-0.154032000
H	6.743088000	1.362499000	-0.798613000
C	4.104094000	0.305139000	2.513013000
C	2.861699000	0.469006000	3.128550000
C	4.784745000	-0.908130000	2.633174000
H	2.338341000	1.410785000	3.005483000
H	5.737869000	-1.019540000	2.129155000
C	2.312413000	-0.571167000	3.870808000
C	4.229014000	-1.943685000	3.376774000
H	1.345690000	-0.438708000	4.347799000
H	4.762246000	-2.885474000	3.467816000
C	2.993545000	-1.779634000	3.999755000
O	4.679227000	1.344164000	1.843405000
H	2.562248000	-2.590055000	4.579184000

=====
TS4
=====

Charge: 0, Spin: 1

Imaginary frequency: -115.9079 cm⁻¹

Cartesian coordinates:

ATOM	X	Y	Z
C	3.890915000	1.565494000	-2.513579000
C	4.742231000	0.707728000	-3.201097000
C	2.628290000	1.860259000	-3.020299000
H	5.711673000	0.472860000	-2.776586000
H	1.979031000	2.524353000	-2.460207000

C	4.331149000	0.157381000	-4.408929000
C	2.229231000	1.300476000	-4.227483000
H	4.999118000	-0.510671000	-4.946416000
H	1.245055000	1.534897000	-4.624754000
C	3.070692000	0.441721000	-4.940571000
O	4.328517000	2.204128000	-1.372792000
O	2.539283000	2.150709000	0.449406000
S	3.800273000	1.491606000	0.135163000
O	3.785204000	0.099465000	-0.309175000
C	2.615179000	-0.188038000	-6.232762000
H	1.946545000	0.476946000	-6.786787000
H	3.462918000	-0.431723000	-6.879129000
H	2.068568000	-1.118445000	-6.041898000
N	6.988047000	4.508603000	0.821009000
C	6.289062000	3.377508000	0.500229000
N	5.007450000	3.277596000	0.617672000
C	4.186807000	4.361963000	1.120727000
C	5.001865000	5.311379000	1.985068000
C	6.268234000	5.704889000	1.243637000
C	8.399220000	4.671565000	0.482246000
C	8.704917000	4.672413000	-1.020282000
C	8.814463000	3.269771000	-1.619703000
C	7.553664000	2.415887000	-1.484743000
C	7.073356000	2.204805000	-0.037968000
H	3.739106000	4.886272000	0.267628000
H	3.364306000	3.919865000	1.684268000
H	5.270662000	4.814568000	2.922925000
H	4.417805000	6.202193000	2.230854000
H	6.927369000	6.286243000	1.895394000
H	6.024612000	6.332697000	0.374012000
H	8.704395000	5.626222000	0.916769000
H	8.998948000	3.902748000	0.985077000
H	7.930249000	5.252699000	-1.536736000
H	9.653331000	5.198373000	-1.178093000
H	9.079656000	3.349933000	-2.679403000
H	9.649913000	2.749502000	-1.130515000
H	7.757251000	1.430808000	-1.917687000
H	6.729605000	2.843318000	-2.065169000
H	7.926180000	2.004581000	0.621406000
H	6.435179000	1.325544000	0.011187000
C	4.363230000	-0.005342000	2.262951000
C	3.123069000	-0.313083000	2.819205000
C	5.432224000	-0.888812000	2.371994000
H	2.305979000	0.391519000	2.707934000
H	6.382930000	-0.621522000	1.922908000
C	2.960024000	-1.518326000	3.491039000
C	5.262434000	-2.089108000	3.054519000
H	1.995398000	-1.765185000	3.923482000
H	6.095180000	-2.779721000	3.145161000
C	4.027231000	-2.407145000	3.613221000
O	4.580447000	1.222633000	1.679707000
H	3.894911000	-3.346080000	4.141347000

Reviewers' Comments:

Reviewer #1:

Remarks to the Author:

The authors have addressed all of my comments, and i support publication of this manuscript in Nature Comm.

Reviewer #2:

Remarks to the Author:

The authors strengthened an already very nice paper!

The correction of minor errors (like the dispersity), the addition of quantum chemistry to clarify the mechanism, and their solid comments to both my comments and those of the other two reviewers make that I think the paper is acceptable in its current form. Congrats to the authors for a beautiful piece of work!

Manuscript Type: Research Article

Title: Iterative SuFEx approach for sequence-regulated oligosulfates and its extension to periodic copolymers

REVIEWER COMMENTS

Reviewer #1 (Remarks to the Author):

The authors have addressed all of my comments, and i support publication of this manuscript in Nature Comm.

Response) We sincerely appreciate the reviewer's valuable comments.

Reviewer #2 (Remarks to the Author):

The authors strengthened an already very nice paper!

The correction of minor errors (like the dispersity), the addition of quantum chemistry to clarify the mechanism, and their solid comments to both my comments and those of the other two reviewers make that I think the paper is acceptable in its current form. Congrats to the authors for a beautiful piece of work!

Response) While addressing the reviewer's constructive feedback, the revised manuscript could be improved. We deeply appreciate the comments provided by the reviewer.

Editorial Requests

Thank you for submitting your manuscript "Iterative SuFEx approach for sequence-regulated oligosulfates and its extension to periodic copolymers" to Nature Communications. I am delighted to say that we are happy, in principle, to publish it under the open access CC BY license (Creative Commons Attribution 4.0 International License).

First, we ask you to revise your paper one last time to address our editorial requests (in the attached author checklist) and any remaining comments from reviewers (included at the end of this email, if applicable).

Failure to comply with our editorial requests will cause delays in accepting your manuscript. Please

also see the Nature Communications formatting instructions for further information.

Response) The authors sincerely appreciate the editorial comments. We have incorporated the detailed responses regarding the editorial requests into the Author Checklist file